# Dengue Virus NS4b N-Terminus Disordered Region Interacts with NS3 Helicase C-Terminal Subdomain to Enhance Helicase Activity

**DOI:** 10.3390/v14081712

**Published:** 2022-08-03

**Authors:** Satyamurthy Kundharapu, Tirumala Kumar Chowdary

**Affiliations:** School of Biological Sciences, National Institute of Science Education and Research, An OCC of Homi Bhabha National Institute, Bhubaneswar 752050, India; k.satyamurthy@niser.ac.in

**Keywords:** flavivirus, dengue virus (DENV), plus-stranded RNA virus, viral replication, RNA helicase, DENV NS3 helicase-cum-protease, SF2 helicase, subdomain 3, DENV NS4b protein, protein–protein interaction, enzyme mechanism

## Abstract

Dengue virus replicates its single-stranded RNA genome in membrane-bound complexes formed on the endoplasmic reticulum, where viral non-structural proteins (NS) and RNA co-localize. The NS proteins interact with one another and with the host proteins. The interaction of the viral helicase and protease, NS3, with the RNA-dependent RNA polymerase, NS5, and NS4b proteins is critical for replication. In vitro, NS3 helicase activity is enhanced by interaction with NS4b. We characterized the interaction between NS3 and NS4b and explained a possible mechanism for helicase activity modulation by NS4b. Our bacterial two-hybrid assay results showed that the N-terminal 57 residues region of NS4b is enough to interact with NS3. The molecular docking of the predicted NS4b structure onto the NS3 structure revealed that the N-terminal disordered region of NS4b wraps around the C-terminal subdomain (CTD) of the helicase. Further, NS3 helicase activity is enhanced upon interaction with NS4b. Molecular dynamics simulations on the NS4b-docked NS3 crystal structure and intrinsic tryptophan fluorescence studies suggest that the interaction results in NS3 CTD domain motions. Based on the interpretation of our results in light of the mechanism explained for NS3 helicase, NS4b–NS3 interaction modulating CTD dynamics is a plausible explanation for the helicase activity enhancement.

## 1. Introduction

Dengue virus (DENV) is a mosquito-transmitted flavivirus. DENV infection causes dengue fever. Severe forms of the disease, either dengue haemorrhagic fever or dengue shock syndrome, can be fatal. Nearly one-third of the world’s population living in the tropics and the subtropics is at risk of dengue [1,2]. The disease is endemic to several southeast Asian countries, including India (an estimated 13 million cases from 2017 to 2018) [3]. Currently, Dengvaxia (developed by Sanofi Pasteur) is the only licensed vaccine recommended for use in high-disease-prevalence regions. However, the vaccine efficacy against serotypes 1 and 2 is only ~50%, as per the phase III clinical trials [4]. Searching for potential targets for antiviral drug development is essential to managing the disease.

The dengue viral genome is a positive-sense single-stranded RNA of nearly 11,000 bases. The viral RNA codes (as a single ORF) for a single, long polypeptide in the infected cell, which is proteolytically processed into non-structural proteins (NS): NS1, NS2a, NS2b, NS3, NS4a, NS4b, and NS5 and three structural proteins. Of these seven NSPs, enzymatic functions in the viral genome replication are known only for NS3 (a protease and helicase) and NS5 (an RNA-dependent RNA polymerase). The coding region of the viral genome is flanked by 5′- and 3′-untranslated regions (UTRs) (Figure 1A). The viral RNA also serves as a template for negative-sense RNA strand synthesis during replication. The negative-strand synthesis starts from the 3’-end of the genome. Translation versus replication of the viral RNA is decided by the cyclization of the genome—a cyclized genome will allow replication but not translation [5,6,7]. Secondary structures of the RNA in the 5’- and 3’-UTRs regulate genome cyclization and promoter selection for negative-strand synthesis [5]. The viral RNA is cyclized through base pairing between conserved complementary sequences in the 5′- and 3′-UTR (Figure 1A). The viral RNA-dependent RNA polymerase (RdRp), NS5, recognizes and binds the conserved stem-loop structure, SLA, in the 5’-UTR and then translocates onto the 3’-end of the circularized genome (Figure 1A) to begin negative-strand synthesis [5]. Interestingly, NS5 cannot bind to the 3′- stem-loop (SL) in the 3′-UTR [6] and cannot proceed with replication if the 3′-SL is not resolved by a helicase [8]. The viral helicase, NS3, through its interaction with NS5, is the most probable interacting protein partner of NS5 to load it onto the 3’-end of the genome. NS3 unwinds the 3’-UTR secondary structures before NS5 can proceed with negative-strand synthesis. The unwinding of the duplex regions in the cyclized genome is an essential step in viral protein synthesis and genome replication [9,10].

In the infected cell, the NS proteins, viral RNA, and RNA duplex intermediates of replication (Replication Complexes, RC) localize to specialized membrane-bound complexes on the ER membranes of various morphologies, collectively defined as replication organelles (RO) [11,12,13,14]. As the viral polyprotein is synthesized, stretches of the polyprotein (corresponding to NS2a, 2b, 4a, 4b, and a small peptide, 2K, connecting 4A and 4B) insert into the ER membrane (bottom row Figure 1A). The viral proteins, especially NS4a and NS4b, and other cellular proteins such as reticulon RTN3.1A [15] and TMEM41B [16] induce curvature on the ER membrane. A predicted topology model of the RO shows that parts of these proteins (loops connecting TM helices) are located in the ER lumen side of the RO (Figure 1A). NS3 and NS5 proteins, on the other hand, reside in the RO lumen, whose contents are in a continuum with the cytoplasm (Figure 1A). However, NS3 is tethered to the membrane surface, from the cytosol side of the RO, through interactions with NS2b, NS4b, and NS4a. Apart from NS3 and NS5, other NS proteins are also required for replication, although enzymatic functions are not known. NS4b is also essential for replication, as mutations in NS4b result in defects in replication and packaging [17,18,19]. 

NS3 is a multidomain multi-function protein. The N-terminal one-third part of the 619 amino acid long protein (in DENV serotype 1) folds into an independent domain with a canonical chymotrypsin-like fold. This domain, along with the NS2b protein co-factor, acts as a serine protease in the proteolytic processing of the polyprotein. The C-terminal reminder of the protein folds into three distinct structural subdomains (Figure 1B) and together functions as an RNA helicase. The N-terminal protease domain is connected to the helicase domain by a flexible linker. The crystal structure of the helicase domain alone [20] or full-length DENV NS3 [21,22] is published. The subdomains 1 and 2 of the helicase have a Rec-A-like fold (RecA-1 and RecA-2 in Figure 1B) and have the canonical sequence motifs (I, Ia, II, III, IV, V, VI) of a p-loop containing nucleoside triphosphatase (ATPase activity) and RNA helicase. The C-terminal subdomain (CTD) is seen in all flavivirus NS3 proteins. Based on the structural features and canonical helicase sequence motifs, NS3 is categorized into the DExH subfamily RNA helicases of the SF2 superfamily of monomeric helicases [23]. This family of helicases load onto a free 3’-end of the polynucleotide and unwind the duplex in the 3’- to 5’- direction.

Different ligand-bound structures of DENV NS3 helicase [23] show that an ATP molecule binds in the interface of the two RecA domains. (Figure 1B), whereas an ssRNA binds in the cleft formed in the interface of the RecA domains and the CTD (Figure 1B). A structural mechanism of ATP hydrolysis-coupled RNA unwinding is proposed [23]. The RNA binding and ATP hydrolysis lead to the inward movement of the p-loop (motif Ia), a twisting motion of the RecA domains on each other, and the outward movement of CTD away from the RecA domains. These domain motions, especially RecA domain motions, lead to the translocation of the RNA in the RNA-binding cleft coupled to every cycle of ATP hydrolysis. However, several aspects of the helicase mechanism are not understood completely. For example, the role of the CTD subdomain in the helicase mechanism is not understood well. The CTD is involved in interactions with other NS proteins, NS4b [18,24] and NS5 [25]. In other SF2 superfamily members wherein subdomains analogous to the CTD are present (domains additional to Rec A-like domains), those subdomains are used for interactions with other proteins that act as regulators of the helicase function [26,27]. Several experimental pieces of evidence [18,24] support the idea that NS3–NS4b interaction is critical for viral replication. 

Interestingly, DENV NS3 helicase, other flavivirus helicases, and a few other superfamily members show very poor RNA helicase activity without their interacting proteins [27,28]. The in vitro helicase activity of DENV NS3 helicase is enhanced upon interaction with NS4b protein [28]. However, the mechanism of the helicase activity modulation is not known. 

NS4b is a predicted transmembrane protein. NS4b co-localizes along with NS3 in replication complexes [11]. Studies by Miller et al. (2006) [29] provided the first biochemical evidence for two helices in the N-terminus of NS4b that peripherally associate with the membrane (named TM1 and TM2) and three transmembrane helices [spanning regions 101−129(TM3), 165−190(TM4), and 217−244(TM5) in DENV serotype 2 NS4b sequence] in the C-terminal half of the protein. As per this proposed membrane topology, the N-terminal (until the beginning of the TM3) region of the protein would be on the ER lumen side, and the loop connecting the TM3 and the TM4 (cytosolic loop) is the only possible interaction site with the NS3. Following this idea, Zou et al. (2015) [24] and Chatel-Chaix et al. (2015) [18] independently studied interaction specifically between the NS4b cytosolic loop and NS3 to fine-map the interacting residues. However, these studies did not rule out the possibility that other regions of NS4b also interact with NS3. Contrary to the above-cited studies, Lu et al. (2021) [30] showed the N-terminal region of NS4b, but not the cytosolic loop, to be an important determinant of interaction with NS3. 

Interestingly, the N-terminal half of NS4b (spanning the TM1 and TM2 helices) plays a role in modulating the innate immune response by inhibiting IFNα/β signaling, probably through direct interaction with STAT1, and modulating its nuclear localization [31]. Since STAT1 is a cytosolic protein, it is reasonable to think that the N-terminal 100 amino acids of NS4b are localized on the cytosolic side of the RO (as depicted in Figure 1B)—in contrast to the proposed topology for NS4b by Miller et al. [29]. Thus, it is possible that the N-terminal region of NS4b is an interacting site for NS3. For our study, we favor the idea that the N-terminal 100 amino acids region of NS4b is on the cytosolic side of the RO (as depicted in Figure 1B). The Section 3 and Section 4 present other reasons for making this assumption. 

As NS3 and NS4b proteins and their interaction are critical for viral replication, they are potential drug targets [17,32,33,34,35]. Recently, a small-molecule drug, JNJ-A07, that blocks NS3–NS4b interaction is shown to be a highly effective pan-serotype DENV inhibitor [35]. A complete understanding of the NS4b–NS3 interaction interface and the mechanism by which NS4b can modulate the helicase enzyme activity of NS3 will greatly help in devising such drug strategies. Our study thoroughly characterized the NS4b–NS3 interaction and provided a map of the NS4b–NS3-interaction interface. Our studies were also aimed at understanding the structural mechanism of the helicase activity enhancement upon NS3 interaction with NS4b. Upon interaction with NS4b, there are significant tertiary structure changes in NS3. Based on these observations, we propose a plausible mechanism for the NS4b interaction-dependent enhancement of NS3 helicase activity wherein the interaction between the proteins modulates the dynamics of the CTD domain motion towards and away from the RNA-binding cleft, leading to helicase activity enhancement. 


Figure 1(**A**) Schematics of the flavivirus genome and replication organelle organization and negative-strand synthesis mechanism. Top: genome organization of flaviviruses: different secondary structures in the 5′-untranslated region (UTR) and 3′-UTR are shown, along with the protein-coding region. The 5′-UTR (stem-loop A—SLA, (U)_n_—PolyU stretch, stem-loop B-SLB, cHP-capsid-hairpin) and the 3′-UTR (the conserved domain III stem-loop—3′-SL, short hairpin—sHP, dumbbell structures in domain III-DB) structures are depicted. The blue boxes labeled 5′-CL and 3′-CL are 5′- and 3′-complementary sequences, respectively. The coding region of the genome is shown as connected boxes of different color. The region that codes for non-structural proteins is labeled. Middle: schematic view of the circularized genome. Bottom: topology of a flavivirus replication organelle (RO) [11,12,13,14,15]. The topology of the different non-structural proteins in the RO is shown. NS2a, part of NS2b, NS4a, and NS4b—membrane-spanning proteins. NS3, circularized viral RNA, and NS5 are shown in the cytosolic side of the RO. The N-terminal tail of NS4b may extend into the cytosolic side of the RO for interacting with cellular proteins and NS3. The NS3–NS5 complex localized to the 3′-end of the genome during negative-strand synthesis are depicted. The zoomed-in section shows a possible scenario of NS3 unwinding the 3′-SL and NS5 trailing during negative-strand synthesis; (**B**) schematic of the domain organization of the DENV serotype 1 NS3 protein. The boundaries of each domain truncation used in the study are marked. RecA 1 and RecA 2 are Rec-A-like subdomains in the helicase domain. CTD is a C-terminal subdomain of the helicase. A cartoon with a transparent surface representation of the DENV serotype 2 NS3 structure (PDB: 2WHX) is shown below, with each domain and different SF2 superfamily helicase sequence motifs marked on it. ATP-binding (marked NTP-binding) and RNA-binding sites are marked. The NS3 structure figure is made using WebGL NGLviewer. Below, a predicted membrane topology of NS4b is shown. The boundaries of transmembrane helices are marked as per the model proposed by Miller et al. (2006) [29], except that the orientation of the N-terminus and C-terminus are flipped to the opposite side, assuming that the N-terminus of the protein is in the cytosol side. As a result, the TM3–TM4 loop (‘cytosolic loop’) is on the ER lumen side. The topology figure was prepared using TexTopo package in LaTex software; (**C**) images of colony patches from a bacterial two-hybrid assay to test NS3–NS4b interaction. A representative image of three replicates from each experiment are shown.
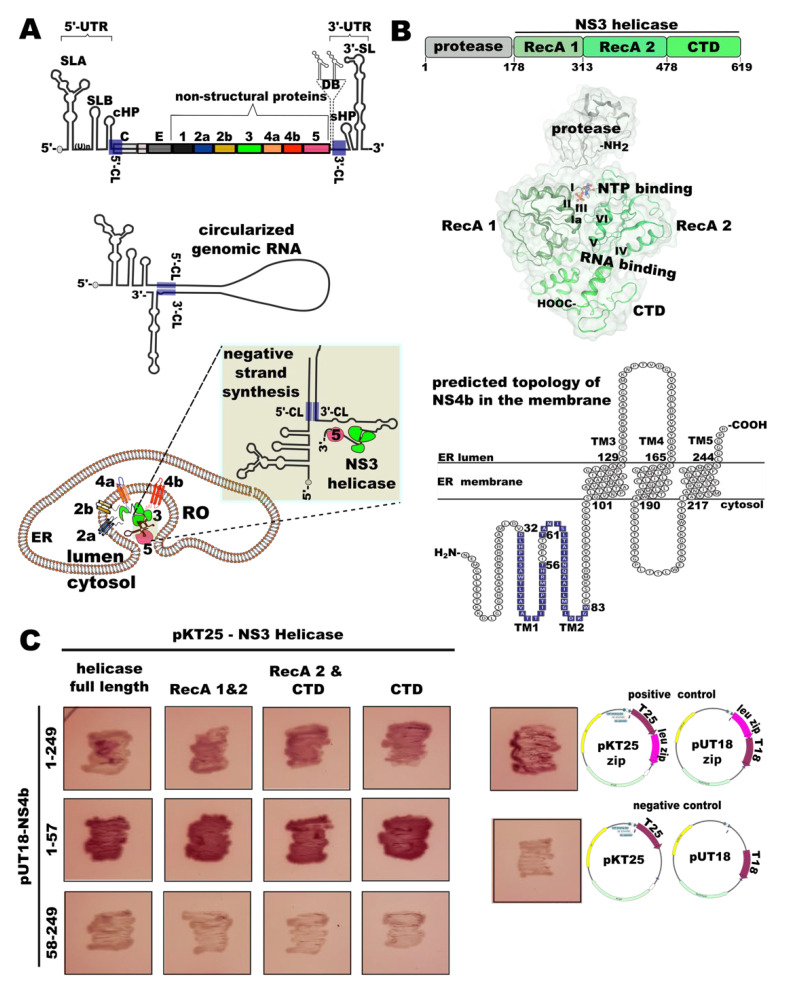



## 2. Materials and Methods

### 2.1. Dengue Virus Sequences and Sequence Predictions 

The sequences of the non-structural proteins of the dengue virus used in this study are from NCBI GenBank accession number JN903579. Transmembrane helix predictions on the sequences are done with TMHMM, Phobius, or PSIPRED MEMSAT-SVM by submitting the NS4b sequence on the respective servers. 

To check if there are any conserved secondary structures in the 3′-UTRs of flavivirus genomes, we first analyzed the last 120 nucleotide sequence of the DENV serotype 1 genome (corresponding to the domain III region of the 3′-UTR). We performed a multiple sequence alignment of this sequence with other flaviviruses’ corresponding 3′-UTR sequences. 

To predict secondary structures in the DENV 1 3′-UTR sequence, we used RNAstructure application on the web server at the URL: https://rna.urmc.rochester.edu/RNAstructure.html (accessed on 7 July 2017). We did RNA secondary structure alignment with 14 different flavivirus sequences (including DENV serotypes 1, 2, 3, and 4 3′-UTR sequences) on the RNAalifold server (http://rna.tbi.univie.ac.at/; accessed on 12 July 2017). The RNA and DNA stem-loop forming oligos that we used in the helicase assays (see below) were designed based on the secondary structure prediction in the RNAstructure web application. 

### 2.2. Reagents

Isopropyl β-D-thiogalactoside (IPTG), ampicillin, kanamycin, ethidium bromide, and guanidine hydrochloride (Gd.HCl) were procured from Sisco Research Laboratories, India. MacConkey agar (+maltose) was from BD, USA. N-Lauroyldimethyl amine oxide (LDAO) was from Merck, Germany. 1-palmitoyl-2-oleoyl-glycerol-3-phosphocholine (POPC) and cholesterol were from Avanti polar lipids, USA. The Superdex-200 10/300 GL increase and the Superdex-75 10/300 GL increase columns were from Cytiva, USA. SYBR Gold (SYBR) stain and ATP were from ThermoScientific, USA. Ni-NTA Agarose and anti-6-histidine tag antibody HRP conjugate were from Qiagen, Germany. The dengue virus NS3 protein antibody was procured from GeneTex, Irvine, CA, USA.

### 2.3. Bacterial Two-Hybrid Assay

A bacterial two-hybrid assay (BACTH) was performed following the protocol described earlier [36]. Full helicase domain (residues 178–619 as per DENV serotype 1 sequence numbering scheme) or only the RecA domains (residues 178–478) or RecA-2 and CTD subdomains (residues 313–619) or only the CTD subdomain of NS3 helicase (residues 478–619) were cloned as a C-terminal fusion with the T25 fragment of the adenylate cyclase of the BACTH assay system, in the pKT25 plasmid. The resulting plasmids are named pSM61, pSM63, pSM64, and pSM62, respectively. Similarly, coding DNA of full-length NS4b_1-249_ (pSM55) or N-terminal 57 residues (plasmid pSM59) or NS4b residues 58–249 (plasmid pSM60) from DENV serotype 1 is cloned as an N-terminal fusion of the T18 fragment of the adenylate cyclase in the pUT18 vector. To perform the BACTH assay, different combinations of pKT25-NS3 helicase and pUT18-NS4b plasmids were used for transforming the *E. coli* BTH101 strain. The transformant cells were plated on a LB agar plate containing 50 µg/mL kanamycin and 100 µg/mL ampicillin. The transformants that grew on the plate were patch streaked on a MacConkey-maltose agar indicator plate supplemented with 50 µg/mL kanamycin, 100 µg/mL ampicillin, and 0.5 mM IPTG and incubated at 30 °C for 96 h. The plasmids pUT18-Zip and pKT25-Zip, containing the interacting domains of the GCN4 leucine zipper, were used as a positive control in the assay. Empty pUT18 and pKT25 plasmids were used as a negative control for the assay. Each experiment was done in triplicates.

### 2.4. Homology Structure Modeling of NS3, Ab-Initio Structure Prediction of NS4b, and Molecular Docking Simulations

We built a homology model of DENV serotype 1 NS3 using DENV serotype 3 NS3 structure (PDB ID: 2WHX) using the SWISS-MODEL server (https://swissmodel.expasy.org; accessed on 13 February 2020). We used two newly released machine-learning algorithm-based protein-structure-prediction software, AlphaFold2.0 [37] and RoseTTAfold [38], for predicting a structure model of DENV serotype 1 NS4b. For AlphaFold2.0 predictions, we used the AlphaFold Colab notebook (shared publicly through a Creative Commons Attribution-NonCommercial 4.0 International license). This AlphaFold Colab notebook uses the BFD database for prediction, after a multiple-sequence alignment. An energy-minimized predicted model PDB file was downloaded along with the pLDDT scores. Structure figures were prepared with PyMOL. We also submitted the DENV serotype1 NS4b sequence on the RoseTTAfold server for structure prediction. The five best structure models, as scored based on angstrom error estimate per residue parameter, were used. Since there was no significant difference in the predictions using AlphaFold2.0 or RoseTTAfold, we used the structure predicted by AlphaFold2.0 for further analysis.

For performing molecular docking simulations, we used the energy-minimized structure of the homology model that we built for DENV serotype 1 NS3 and the predicted structure of NS4b. A molecular docking simulation was performed on the Haddock server [39]. We first did a rigid body, global, blind docking run on the ClusPro server (https://cluspro.bu.edu; accessed on 20 July 2021) to get a possible docking pose and information about the interacting residues. Using the set of residues in the interface of the docked pose from the ClusPro docking run, we set up a Haddock 2.4 (https://wenmr.science.uu.nl/haddock2.4; accessed on 23 July 2021) docking run. The N-terminal 57 residues of NS4b were set as active and flexible regions in the docking run. The docked pose with rank 1 docking score was used for further analysis. 

Molecular dynamic simulations were performed on the NS4b-docked NS3 structure using GROMACS software. We set up two different simulations runs: one with the NS3 structure alone and another with the NS3–NS4b complex structure (corresponding to the best docking pose from the molecular docking studies). Both proteins were solvated with water (water model-TIP3P), and no ions were added. The NS4b-docked NS3 protein model was enclosed in a box of 14.903 nm, 14.910 nm, and 14.906 nm dimensions. The system was energy-minimized in an NVT ensemble and equilibrated for 125 picoseconds in the NPT ensemble. Each system was simulated using Gromacs-5.1 with a CHARMM36 forcefield with a 1 fs step time. Periodic boundary conditions were applied in the simulation. Following equilibration, the system was subjected to the production of simulation for 20 ns at 303 K.

### 2.5. Bacterial Expression and the Purification of NS3 and NS4b Proteins

We expressed the helicase domain of DENV NS3 (NS3Hel) in an *E. coli* expression system and purified the protein using six-histidine-tag affinity chromatography, following the protocol given by Xu et al. (2005) [20] with a few modifications. DNA clones with NS3 and NS4b coding regions, pDV1-419NS2b-3 and pDV1-419NSP4b, respectively, were a donation from Dr. E. Sreekumar, Rajiv Gandhi Centre for Biotechnology, Thiruvananthapuram. The coding DNA corresponding to NS3 residues 178 to 619 was PCR-amplified with forward (5′-CCAGCTAGCGAGATTGAGGACGAGGTG-3′) and reverse primer (5′-ACTCTCGAGTCTTCTTCCTGCTGCGAA-3′) and cloned into pET24b vector using Nhe I and Xho I sites (pSM10) so that the resultant protein has a 6-histidine-purification tag as a C-terminal fusion. The coding region of DNA corresponding to NS4b residues 1–249 was PCR-amplified using forward and reverser primers (5′-AGACATATGAATGAGATGGGATTA CTGGAAACCACAAAG-3′ and 5′-ACCCTCGAGTCTCCTACCTCCTCCCAA-3′), respectively, and cloned into pET24b with a C-terminal six-histidine tag (pSM6). All clones were sequence verified. 

For expression, competent cells of Rosetta (DE3) *E. coli* were transformed with pSM10 plasmid. Transformant cultures were grown in LB Broth supplemented with 50 µg/mL kanamycin and 34 µg/mL chloramphenicol to an optical density at 600 nm (OD_600_) of 0.6. Protein expression was induced by 0.3 mM isopropyl β-D-1thiogalactopyranoside (IPTG), after which cultures were grown at 14 °C for 20 h. Cells were harvested by centrifugation and lysed by sonication in 20 mM Tris-Cl, pH 7.4, buffer containing 1M NaCl. Protein purification was done from the clarified lysates using 6-His-tag affinity to a Ni-NTA agarose column. The protein was bound to Ni-NTA with 20 mM imidazole containing buffer, followed by a 50 mM imidazole wash step and elution with 300 mM imidazole containing buffer. As a final purification step, the 300 mM imidazole elution fraction from the His-tag-affinity purification step was concentrated and loaded on to a Superdex-75 Increase 10/300 GL gel-filtration column. Elution was done with 20 mM Tris-Cl, pH 7.4, buffer containing 150 mM NaCl and 1 mM EDTA. The purity of the protein was assessed by an SDS-PAGE analysis of the concentrated fraction of the gel-filtration chromatography eluent. Protein concentration was estimated by recording absorbance at 280 nm and using the E1% value of 13.54. 

For the expression and purification of the NS4b full-length, we adapted the bacterial expression and protein detergent extraction protocol published by Zou et al. (2014) [19]. The plasmid pSM6 was used to transform BL21(DE3) *E. coli*. The transformant bacterial culture was grown for 16 h at 18 °C, after protein expression induction with 0.5 mM IPTG. The cells were harvested by centrifugation and lysed by sonication in 20 mM Tri-Cl, pH 7.4, containing 150 mM NaCl (TN). The NS4b was in the pellet fraction after the clarification step of centrifugation at 30,000× *g* for 30 min. The protein was solubilized and extracted out of the pellet by detergent extraction protocol using incubation in TN buffer containing 3% LDAO and 3 M guanidine hydrochloride for 1 h at RT. After a centrifugation clarification step (30,000× *g* for 30 min), the clear extracts were diluted to reduce the concentration of the detergent and Gd.HCl to 1% and 1 M, respectively, before loading onto a Ni-NTA agarose column. The protein was eluted as a pure fraction with 200 mM imidazole and 0.1% LDAO containing TN buffer. The fractions containing the pure NS4b protein, as assessed by SDS-PAGE analysis, were concentrated. The concentrated fraction was loaded on to a Superdex-75 increase 10/300 GL gel-filtration column and eluted with TN buffer containing 0.05% LDAO and 1 mM EDTA. NS4b protein eluted at different elution volumes in the gel-filtration chromatography. Each elution fraction was collected separately and concentrated by ultra-filtration. Protein concentration was estimated by recording Abs._280 nm_ and using E1% value of 12.6. 

We also purified NS4b without using a detergent extraction protocol. The protein expression conditions differed from the above-described protocol in that the incubation temperature was set to 37 °C and protein expression induction was done by adding 1 mM IPTG to the culture for 4 h. The cell lysis was done by sonication and lysates were clarified using centrifugation. The pellet fraction after the clarification step was incubated with 6 M Gd.HCl containing TN buffer overnight. After a clarification step by centrifugation, the clear fractions from the denaturant extraction step were loaded onto a Ni-NTA agarose column. The protein was slowly refolded on the column by changing the concentration of Gd.HCl in the wash and elution steps sequentially to 3 M and 1 M. The protein thus eluted off the column with 300 mM imidazole and 1 M Gd.HCl containing TN buffer was further diluted, so that the concentration of the denaturant was reduced to 0.05 M. The protein was kept on ice at this step. Subsequent purification steps included the concentration of the protein using ultrafiltration and then gel-filtration chromatography on a Superdex-75 column. The protein thus obtained is more than 95% pure. 

The N-terminal 57 residues region of NS4b (NS4bN57) was expressed in a similar way. The protein was expressed from a pET24b vector in BL21(DE3) *E. coli* cells. The expression conditions were: culture growth temperature set to 30 °C and protein expression induction with 0.5 mM IPTG. The NS4bN57 expressed in the form of a soluble protein. The protein was purified from the clarified lysate (in TN buffer) using Ni-NTA His-tag affinity chromatography. 

### 2.6. Far-UV Circular Dichroism (CD) and Intrinsic Tryptophan Fluorescence Spectroscopy

Far-UV CD measurements were done using a Jasco 1500 spectropolarimeter. The protein concentration was 0.2 mg/mL in each case. Spectra were recorded at 50 nm/min scan speed in the wavelength range 200–250 nm. Bandwidth was set to 1 nm. Each spectrum was an accumulation of three scans. Each spectrum was blank-corrected (TN buffer alone or TN buffer along with 0.05% LDAO when NS4b purified by detergent extraction method was used) and smoothened. Mean residue ellipticity, θ_m.r.e_, was calculated using the equation [θ]_m.r.e_ = 100 × θ/(CNl), where θ is the observed ellipticity at a given concentration of protein, C is the molar concentration of the protein, N is the number of residues in the protein, and l is the cell path-length in centimeters. 

For studying secondary structural changes in NS3Hel or NS4b or both upon interaction, we followed the method described by Greenfield NJ, 2015 [40]. The far-UV CD spectra were recorded for NS3Hel alone or NS3Hel:NS4b protein mixtures at different molar stoichiometries (1:1, 1:2, 1:3). The NS4b protein that we used in this assay was purified following the detergent-free protocol described above. The data was analyzed by converting the ellipticity values to θ_m.r.e_. In the mixture of the two proteins, the θ_m.r.e_ was calculated using the cumulative molecular weight of the monomer mass of NS3 and the monomer mass of NS4b (51.8 kDa for NS3Hel + 27.8 kDa for NS4b). If there were any secondary structure changes, upon interaction, it was expected that the observed θ_m.r.e_ for the mixture would be different than the expected value.

Intrinsic tryptophan fluorescence spectra were recorded using a FLS 1000 fluorimeter (Edinburgh Instruments, Livingston, UK). The protein was used at a 0.05 mg/mL concentration for recording the fluorescence spectra. The excitation wavelength was set to 295 nm, and the emission wavelength range was set to 300–500 nm. The excitation and emission bandwidths are set to 1 nm. Spectra were blank-corrected and smoothened using the Fluoracle (Edinburgh Instruments, Livingston, UK) software. For each spectrum recorded on the NS3:NS4b mixture, the spectrum of the corresponding NS4b concentration was subtracted, following the protocol described earlier [41].

### 2.7. Bio-Layer Interferometry (BLI)

To test the interaction between NS3Hel and NS4b proteins, we used a Bio-layer interferometry assay. Measurements were recorded using the Octet (FortéBio) platform using High Precision Streptavidin (SAX) Biosensors (FortéBio) (courtesy of FortéBio). Using standard reagents and protocols supplied by the manufacturer (FortéBio), the NS4b protein was biotinylated. The NS4b protein was loaded onto the sensor at 2 μg/mL concentration. After a wash step with BLI buffer, the NS4b-bound sensor was incubated with different concentration of NS3Hel, (20, 6.647, 0.7407 μM) in TNE with 0.05% tween-20. Binding was followed for 200 s where it reached equilibrium. The biosensor then was incubated in buffer for 200 s to measure the dissociation reaction. The shift in the wavelength of light reflecting from the sensor (in nm) is measured in real-time during the binding and dissociation phases. The binding response curves (sensograms) for the indicated NS3 helicase protein concentrations, over the NS4b-immobilized sensor, are plotted. The equilibrium dissociation constant (K_D_) of binding was estimated from the curve fitting on the response curves using Octet software (FortéBio). 

### 2.8. Liposome Preparation and Co-Floatation Experiments

Large unilamellar vesicles (LUVs) were prepared with synthetic purified lipids POPC and cholesterol in a 1:1 ratio by extrusion method following a standard protocol. A liposome co-flotation assay was performed using NS4b purified using a detergent extraction protocol (200 μg) and 0.5 mM of lipid vesicles in a final volume of 200 μL TNE buffer. Briefly, the liposome–NS4b protein mixture was placed above a 40% sucrose bed (4 mL), which was topped by two layers of 20% and 5% (4 mL each) sucrose. After ultracentrifugation for 3 h at 200,000× *g*, fractions from the top (4 mL) and bottom (4 mL) were collected. Aliquots from the top and bottom fractions in each assay were analyzed with Western blotting using anti-His-tag monoclonal antibodies.

### 2.9. Electrophoretic Mobility Shift Assays for Helicase Activity Measurement

We adapted the protocol described by Xu et al. (2005) [20] for helicase activity measurement using a electrophoretic mobility shift assay (EMSA). We further optimized the protocol with an RNA and a DNA stem-loop oligos that we designed (see dengue virus sequences and sequence predictions section above) to mimic the 3′-SL of the DENV. Briefly, an RNA oligo with the sequence 5′-UCUACAGCAUCAUUCCAGGCACAGAACGCCAAAAAAUGGAAUGGUGCUGUUGAAUCAACAGGUUCUUUUU-3′ with FAM (6-Carboxyfluorescein) fluorophore attached at the 5′- end of the oligo was procured from GenScript, USA. The RNA oligo was visualized after the EMSA gel run using the FAM fluorescence. We also synthesized a 66-nucleotide DNA oligo with sequence 5′-TCTACAGCATCATTCCAGGCACAGAACGCCAAAAAATGGAATGGTGCTGTTGAATCAACAGGTTCT-3′. Before using the oligonucleotides in the EMSA assays, we included a 94 °C denaturation and slow annealing step to ensure a proper stem-loop structure. The helicase assay was done by mixing the RNA (0.75 picomoles per reaction) or the DNA oligo (3 picomoles per reaction) with 20 and 50 molar excess of NS3Hel protein alone, respectively, or with NS3Hel: NS4b in different (1:1, 1:2, 1:3) molar ratios. The final reaction volume was adjusted to 10 μL to achieve a final buffer concentration of 20 mM Tris-Cl, pH 7.4, containing 60 mM NaCl and 1 mM MgCl_2_. The assay was started by adding ATP to the reaction mixture at 2 mM concentration for RNA oligo and 4 mM for the DNA oligo reaction mixture. The reaction was stopped by adding the EMSA loading dye (containing 5% glycerol and 10 mM EDTA) to the reaction mixture at the indicated time points. EMSA was performed on 15% native Tris Borate EDTA (TBE)-polyacrylamide gel. We first confirmed that the 3′-SL DNA/RNA oligos actually form a stem-loop structure by analyzing the oligos on EMSA along with the heat-denatured sample of the oligo in the next lane. The mobility of the heat-denatured oligo in EMSA matched that of unwound stem-loop/duplex substrates used for the DENV NS3 helicase by others [20]. The oligo samples that were not denatured showed slower mobility in EMSA compared to denatured/unwound oligo, confirming that the DNA/RNA oligos formed stem-loop structures as predicted. The RNA or the DNA oligo alone in helicase assay buffer served as negative controls for the respective experiments. We also included NS4b alone and NS3 without adding ATP along with the RNA and the DNA oligo as controls to check that the unwinding activity is not because of some unknown factors coming from the protein preparations or buffer components. After the electrophoretic run staining for DNA was done using SYBR Gold as per the manufacturer’s protocol, and the gel images were recorded. For experiments with RNA oligos, the gels were imaged directly after the electrophoretic run using the Alexa 488 filter of a gel imager. The helicase activity was estimated by taking pixel intensity (from the images of the stained EMSA gels) of the ssRNA (unwound RNA stem-loop) or ssDNA band over the cumulative intensity of the single-stranded and the stem-loop bands. For comparing helicase activity in the absence and in the presence of NS4b, we normalized the helicase activity in each case to helicase activity with NS3 alone. The % helicase activity in each case is plotted in a bar plot along with ± standard error. The average helicase activity with NS3Hel alone at one hour past ATP addition to the reaction mixture was taken as 100%. 

We also assessed NS3Hel RNA duplex unwinding activity using a molecular beacon assay published earlier [42]. Two RNA oligos: CY5- 5′-GACGUCAGUUGUUAGUCUACGUC -3′—BHQ2 (wherein CY5 is a fluorophore with excitation and emission maximum at 630 and 670 nm, respectively, and BHQ2 is a black hole quencher ) and 5′-AGACUAACAACUGACGUCUUUUUUUUUUUUUUUUUUUU-3′, with complementary sequences (underlined text in both sequences), are used to form an RNA duplex. The helicase reaction mixture contained 30 nM of dsRNA with 100 nM of NS3Hel protein alone or NS3Hel: NS4B/NS4BN57 in a 1:3 molar ratio. The final reaction volume was adjusted to 25 μL to achieve a final buffer concentration of 20 mM Tris-Cl, pH 7.4, containing 15 mM NaCl and 2 mM MgCl_2_. The assay was started by adding ATP to the reaction mixture at 2 mM concentration, and fluorescence intensity of CY5 (at 670 nm) was recorded for the next 30 min. Fluorescence intensity (FAU at 670 nm) before the addition of ATP was taken as the starting fluorescence value (F_0_) and fluorescence intensity at 30 min after starting the assay as the assay end point value (F_30_). As the fluorescence of CY5 would be quenched by the black hole quencher after the duplex is unwound and the labeled oligo formed a stem-loop structure, the F_30_ value would be lower than F_0._ The F_0_-F_30_ was taken as 100% helicase activity of NS3Hel. The percent increase in helicase activity in the presence of NS4b or NS4bN57 was calculated using the formula (F’_0_-F’_30_) × 100/(F_0_-F_30_), where F’_30_ and F’_0_ are endpoint and initial fluorescence values in reaction mixtures with NS3:NS4b or NS4bN57 in a 1:3 molar ratio.

### 2.10. ATPase Assay

The ATPase activity of NS3Hel was measured by quantifying the release of free phosphate (P_i_) following ATP hydrolysis by malachite green assay in the presence of a DNA stem-loop following a protocol described earlier [43]. The assay was performed in a final volume of 30 μL by mixing NS3Hel (50 nM) protein alone or NS3Hel:NS4B/NS4BN57 in a 1:3 molar ratio. The NS3 protein was pre-incubated with 0, 0.3, 0.6, 1.2, 2.4, and 4.8 μM of the DNA stem-loop oligo in a reaction buffer of 20 mM Tris–Cl, pH 7.4, 1 mM MgCl_2_, and 60 mM NaCl. The ATPase reaction was started by adding 1 mM of ATP and incubated further for 40 min at 30 °C. The reaction was stopped by adding 20 mM EDTA, and aliquots were collected every 10 min and stored at 4 °C until further processing. A total of 10 μL malachite green reagent (Sigma, USA) was added to a 40 μL sample and incubated at room temperature for 30 min to form a complex with molybdate and free orthophosphate. Samples were transferred into a 96-well plate, and the absorbance at 620 nm was recorded using a Thermo Varioskan microplate reader. From the Abs._620nm_ values, the orthophosphate concentration, and thus the P_i_ released from the ATPase activity of the helicase, was estimated using a calibration curve with known concentrations of orthophosphate. 

For Michaelis–Menten kinetics studies on NS3Hel, the ATPase assay was done with different concentrations of the substrate (stem-loop DNA oligo), keeping the ATP concentration and enzyme concentration in each experiment constant. Initial velocity measurements at different substrate concentrations and fitting the data into a MM-kinetics model was done using ICEKAT web application (http://icekat.herokuapp.com/icekat; accessed on 12 November 2021) [44]. Slopes from the linear range of each malachite green absorbance versus time kinetic trace is taken as the initial rates. The kinetic data is fit to an MM-kinetic model using a non-linear regression method either in the ICEKAT application or in Graphpad Prism software.

## 3. Results

### 3.1. N-Terminus 57 Residues of NS4b Are Enough to Interact with NS3

To fine-map the interaction regions on NS3 and NS4b, we used a bacterial adenylate cyclase-based two-hybrid (BACTH) assay. We tested the interaction of full-length, only the N-terminal 57 residues region, or N-terminal 57 residues truncated NS4b proteins with different subdomains of the NS3 helicase. Our rationale for selecting different regions of NS4b for testing the interaction is as follows: earlier sequence predictions and biochemical studies [29] showed that most of the protein inserts into the ER membrane (TM3, TM4, and TM5) and only the N-terminus 100 residues (or the loop connecting TM3 and TM4, if the protein inserts into the membrane in orientation as proposed by Miller et al. [29]) might be accessible for NS3 interaction. Hence, we were interested to see if the N-terminal region of NS4b alone can interact with NS3. Even in the N-terminal 100 residues, two membrane-associating helices are predicted, although there is inconsistency in transmembrane region predictions reported in different studies. We included only the N-terminus 57 residues (till the end of the predicted first helix) as this region may be the most distal structural element of the NS4b protruding into the cytosol side of the RO and thus most accessible to NS3 for interaction. 

The interaction between the proteins was assessed qualitatively by observing red color development in the transformed bacterial colonies on the MacConkey/maltose agar indicator plates and comparing it with color development on the control plates. As expected, and confirming earlier observations by others [18,24], full-length NS3 helicase and NS4b transformed colonies showed red color development, nearly comparable to that of the positive control bacterial patch (Figure 1C, top row). We observed that RecA-1 and -2 subdomains alone of the NS3 helicase, as well as RecA-2 and CTD and the CTD subdomain alone interacted with NS4b comparably to the full-length NS3 helicase. This implies that the NS3–NS4b interaction interface may spread over a large surface area of NS3, spanning different subdomains of the helicase.

Surprisingly, when only the N-terminus 57 residues region of the NS4b was tested in the assay, the color development and colony growth were much more robust than those observed with full-length protein constructs (Figure 1C, middle row), indicating a more robust interaction strength. We consistently observed a stronger interaction between the N-terminal 57 residues fragment of NS4b with the NS3 helicase full-length protein or with different subdomain variants of it in different replicates of the experiment. On the other hand, the N-terminal-57-residues-deleted NS4b did not show any interaction with NS3 (Figure 1C, bottom row). These observations align with our thinking that the N-terminal end of the NS4b is the only region accessible for NS3 interaction. Supporting this idea, Lu et al. (2021) [30] also noted that the N-terminal region of NS4b (residues 51–83) is enough to interact with NS3. Thus, from our BACTH assays, it is apparent that the N-terminal 57 residues of NS4b (or a sub-region within) is enough for interaction with NS3, and the interaction interface on NS3 spans all three subdomains of the helicase domain. However, from our experiments, we cannot exclude the possibility of other regions, especially the long loops connecting the predicted transmembrane helices, also interacting with NS3. Earlier biochemical studies by Zou et al. (2015) [24] also noted that subdomains 2 and 3 from NS3 helicase are important for interaction with NS4b. 

### 3.2. Molecular Docking Simulation Shows That the N-Terminus Region of NS4b Interacts with RecA-2 and the CTD Subdomains of NS3

To validate our bacterial two-hybrid assay results and to understand the nature of interactions between NS3 helicase and NS4b, we performed a molecular docking simulation with a homology model built for DENV serotype 1 NS3 and predicted the structure model of NS4b. 

The crystal structures of NS3 are published for DENV serotypes 2 and 4. However, no crystal structure is available for the NS3 of DENV serotype 1. We generated a homology model for serotype 1 NS3 using the DENV-serotype two full-length NS3 (PDB Id. 2WHX) structure model as a template. The QMEANDisCo score for the predicted model is 0.81 ± 0.6, and the model has 0.84% Ramachandran outliers.

There is no crystal structure model available for NS4b. Secondary structure prediction on the amino acid sequence predicts seven long helices that span through the majority of the protein sequence, except for the N-terminal 37 residues (Figure 2A). Transmembrane helix predictions using different prediction software are inconsistent. TMHMM did not predict any transmembrane helices, whereas PSIPRED MEMSAT-SVM predicted six transmembrane helices. However, three TM helices are consistently predicted in the regions 100–150, 175–200, and 218–241 by Phobius and PSIPRED MEMSAT-SVM prediction algorithms (Figure 2A). 

To predict the tertiary structure of NS4b, we used two computational protein structure prediction software, RoseTTAfold and AlphaFold 2.0, that use a machine-learning approach and are shown to predict accurately novel protein structures even when a homologous protein structure template is not available. The structure predicted by AlphaFold for DENV1 NS4b is shown in Figure 2B. For most part of the sequence, except for the first 32 amino acid residues, the predicted local distance difference test (pLDDT) score is >90, which implies that the level of confidence of accurate prediction is high for that region (high likelihood of the predicted structure matching with the experimentally derived structure with low Cα-RMSD value). The N-terminus 32 residues region has a pLDDT measure of <70. The N-terminus 36 residues are predicted to be a disordered region, except for a small helix (LLETTKKDL) in the beginning (Figure 2B). The predicted 3D structure of NS4b is in line with the secondary structure predictions on the sequence. We also generated a model for DENV serotype 1 NS3 using AlphaFold2.0, explicitly not using any PDB template. Most of the sequence in the predicted model has a pLDDT score of >90, except for the N-terminal residues of the protease domain and the long loop connecting the protease to the helicase. The structural alignment of the model predicted by homology modeling and the model predicted by AlphaFold 2.0 model (only NS3 helicase domain) showed Cα-RMSD value of 0.886 Å, indicating that the AlphaFold 2.0 prediction is almost as efficient as model prediction by homology modeling. This increased our confidence on the model that we built for NS4b using AlphaFold2.0.

We performed molecular docking simulation with the generated structure models of NS3 and NS4b. The best-docked pose predicted by the Haddock server for the NS3–NS4b complex, as assessed by the lowest intermolecular energy score of −151.8 + 16.6 with a cluster size of 10, is shown in Figure 2C. As expected, only the N-terminal disordered region of NS4b exclusively interacts with NS3. The interaction interface spans RecA-2 and the C-terminal subdomains of NS3. The N-terminus 32 residues disordered region of NS4b runs lateral to the RecA-2 domain, on the side opposite the RNA-binding cleft, and reaches the C-terminus subdomain of NS3. This explains our BACTH results wherein we observed the interaction of both full-length and N-terminal 57 residues region with all NS3 variants that we tested. In the docked position, Lys10 of NS4b and Glu568 in the C-terminal subdomain of NS3 helicase are at a distance to form a salt bridge in between, possibly stabilizing the interaction (Figure 2C, zoomed-in region of the interaction interface). Notably, residues 10 to 28 in the NS4b are positioned next to a loop that connects the RecA2 and CTD in NS3. This loop region exists as a helix in the apo (before ssRNA and ATP binding) form of NS3. This region is proposed to be a hinge (Figure 2C) for the CTD swiveling motion on the RecA-2 after substrate binding [23]. Interestingly, when only the N-terminal 51–83 residues of NS4b were modeled and docked onto the NS3 helicase, the peptide model was positioned between the RecA domains [30].

We also marked the possible boundaries of the membrane (Figure 2C), depicting the possible orientation of the NS4b in the ER membrane. If this were to be the orientation of NS4b TM helices in the ER membrane, then the N-terminal 36 residues disordered region of NS4b would extend as a flexible tail away from the membrane into the cytosol side of the RC. This model is in accordance with the mechanistic model of NS4b–NS3 interaction that we (Figure 1A) and others [16,45] have proposed—only the N-terminus flexible tail of NS4b would extend away from the membrane surface and interacts with NS3. 

### 3.3. Recombinant NS4b and NS3 Proteins Interact Independent of NS4b Membrane Insertion

To further validate our BACTH and molecular docking analysis, we expressed and purified DENV serotype 1 NS3 and NS4b proteins and did a thorough biochemical characterization of the interaction between the proteins.

We cloned only the NS3 helicase domain (as per DENV serotype 1 sequence numbering residues 178–619 of NS3) into a bacterial expression vector in fusion with a C-terminal 6-histidine purification tag. Our bacterial two-hybrid assay results and molecular docking results indicated that only the helicase domain interacts with NS4b. Moreover, earlier studies [28] showed that bacterially expressed helicase domain folds into a functional protein, independent of the protease domain of NS3. The helicase domain of NS3 (NS3Hel) is purified using 6-histidine-tag affinity chromatography (an image of the gel with SDS-PAGE analysis on the purified protein is shown in Figure 3A). The purified protein elutes in gel-filtration chromatography as a single sharp peak at an elution volume that corresponds to the monomer mass of the protein (Figure 3B). Secondary structure analysis by far-UV circular dichroism spectroscopy shows a spectral signature (Figure 3C) typical of a predominantly helical protein, consistent with the earlier reported [23] secondary structure content for the protein.

We expressed and purified the NS4b protein, adopting a protocol that was published earlier. Since NS4b is predicted to be a transmembrane protein, we used LDAO detergent to extract the protein from the insoluble fraction of bacterial cell pellets and in the 6-His Ni-NTA affinity purification protocol. To rule out the possibility of artifacts in the interaction studies due to the detergent, we exchanged the protein into a buffer containing LDAO at critical micellar concentration as the final purification step. SDS-PAGE analysis on the purified protein showed that the protein was more than 95% pure (Figure 3D).

Size-exclusion chromatography on the purified NS4b revealed that the protein forms large oligomers, which are in equilibrium with a dimer and monomer population (Figure 3E). We observed two elution peaks in the resolution range of the gel-filtration column and another broadly spread elution pattern, starting from the void volume and extending into the resolution range. While the protein elution peak at 15.5 mL on a 24 mL Superdex-200 column can be ascribed to the monomer (estimated molecular weight of 27 kDa; A 29 kDa marker protein, carbonic anhydrase, eluted at 16.5 mL), the peak at 14.2 mL corresponds to a dimer molecular mass of NS4b (Figure 3E). The precise oligomer size of the protein, however, could not be determined. All three fractions contained only NS4b and not any contaminant proteins (Figure 3E, inset on the right side). We could not resolve the oligomers even in a wider-resolution-range SEC column, the superose-6 column (data not shown). Furthermore, the equilibrium shifted to large oligomers very rapidly, as the protein formed large oligomers within a few hours of storage at 4 °C (Figure 3E, inset in the left). Interestingly, the oligomers of NS4b are completely soluble, as we did not observe significant precipitation of the protein either on the gel-filtration column or in storage. Similar observations were made by others with purified recombinant NS4b [19] and the protein expressed in DENV-infected cells [28,29]. Far-UV CD spectroscopy (Figure 3F) on the protein reveals that the protein is predominantly helical (negative peaks at 208 nm and 222 nm), as expected from the secondary structure and tertiary structure predictions.

To characterize the interaction between NS3 and NS4b, we wanted to isolate the monomeric fraction of the NS4b protein, as non-specific interaction between the protein molecules in the oligomers may mask the region that interacts with NS3. Despite several attempts and testing different buffer conditions (different non-denaturing detergent concentrations), we could not isolate the monomeric protein. Earlier studies [18] showed that NS4b protein expressed in mammalian cells, after a recombinant dengue virus infection, could be co-immunoprecipitated along with NS3. It is possible that NS4b oligomerization does not affect its interaction with NS3. An alternate possibility is, that NS4b–NS3 interaction may induce conformational changes in the proteins to populate more monomeric NS4b. To test these possibilities, we studied the interaction between the purified NS3Hel and NS4b proteins using BLI. The estimated K_D_ value of the binding equilibrium is 0.508 μM (Figure 3G). These observations suggest that NS4b oligomerization does not affect its interaction with NS3.

To test this further, we did size-exclusion chromatography on mixtures of NS3Hel and NS4b at different stoichiometries (Figure 3H). Fractions collected from each run were analyzed by Western blotting using an NS3-specific polyclonal antibody. Surprisingly, we did not find any interaction between NS3Hel with monomeric NS4b, as there is neither a significant reduction in the NS4b monomeric peak (green dotted lines in Figure 3H) nor the appearance of a new elution peak that would correspond to NS3Hel–NS4b (1:1 molecular ratio) molecular mass, whereas the elution peak corresponding to the NS4b oligomers showed a small but significant increase in absorbance (280 nm) intensity (Figure 3H). When fractions corresponding to this peak were analyzed by Western immunoblotting, a very small amount of NS3Hel could be detected in those fractions, confirming that NS3Hel co-eluted along with NS4b oligomers. In gel-filtration runs with NS3Hel alone, the protein did not elute in those fractions (data not shown).

We then wondered if the oligomerization of NS4b or the membrane insertion of the transmembrane helices of NS4b is a prerequisite for NS3 interaction. It is possible that, upon membrane insertion, NS4b would be presenting the N-terminal 36 residues in the right conformation to interact with NS3, and oligomerization may be providing a similar environment. Prompted by this thinking, we tested the association of NS4b with lipid large unilamellar vesicles. We performed a co-flotation assay with NS4b-lipid LUV mixtures. Contrary to the expectation, we did not find any NS4b association with lipid LUVs (Figure 3I); NS4b is seen only in the bottom fractions. This observation negates our hypothesis that the membrane insertion of NS4b is a requirement for its N-terminus interaction with NS3.

Based on all these results, we conclude that the membrane insertion of NS4b is not necessary for its interaction with NS3 in vitro. A meaningful interpretation of these results is that the N-terminus disordered region of NS4b is enough to interact with NS3, as indicated by our bacterial two-hybrid assays and molecular docking studies. Furthermore, the oligomerization of NS4b does not hinder the protein N-terminal region interaction with NS3.

### 3.4. Interaction between NS4b and NS3 Leads to Conformational Changes in the Helicase Subdomains

To explain the mechanism of NS3 helicase activity modulation by NS4b, we tested if the interaction between NS3 and NS4b leads to any conformational changes in the proteins. We recorded far-UV CD spectra of NS3Hel alone or a NS3Hel–NS4b protein mixture at different stoichiometries. The overlay of the blank-corrected and normalized spectra is shown in Figure 4A. We analyzed the far-UV CD data to interpret the interaction as described by Greenfield NJ, 2015 [40] (see the Section 2 for details). The far-UV CD spectra of NS3Hel and NS4b nearly overlapped. In the normalized spectra at higher ratios of NS3:NS4b, there is no significant change in the spectra of NS3, implying that there are no major secondary structure changes in NS3 helicase upon interaction with NS4b. If there are any secondary structure changes, upon interaction, a significant change in the spectra for the NS3Hel: NS4b mixtures is expected.

Then we wondered if the interaction would result in large domain movements. Such changes in the tertiary structure of the protein can be reported by changes in the intrinsic tryptophan fluorescence. We recorded intrinsic tryptophan fluorescence of NS3Hel in the presence of different molar excess ratios of NS4b and compared it to NS3Hel alone spectrum. As can be seen in Figure 4B, the fluorescence intensity at the emission maximum (*λ*_max emission_: 345 nm) of NS3Hel in the presence of NS4b is high compared to NS3 alone. This signifies that there is a change in the tryptophan environment in NS3 after the interaction. There are nine tryptophan residues in the DENV serotype 1 NS3 helicase domain, of which five are in the C-terminal subdomain (Figure 4C). In the docked pose of NS3–NS4b (from our molecular docking studies), the N-terminal disordered region of NS4b arches around the RecA-2 and C-terminal subdomains. Thus, we interpret the change in tryptophan fluorescence observed in our experiment as due to the N-terminal domain of NS4b interacting with the C-terminal subdomain of NS3 and possibly moving it closer to RecA domains. This would rigidify the environment surrounding the five tryptophans in the CTD, possibly resulting in the intrinsic tryptophan fluorescence changes that we observed. 

Earlier molecular dynamics simulation studies on different flavivirus NS3 structures showed that, upon ATP-binding and hydrolysis, there are allosteric changes in the RNA-binding cleft (between RecA domains and CTD) [46,47]. Based on this, it was explained that the allosteric changes possibly couple the RNA unwinding to ATP hydrolysis. The hinge connecting the CTD to the RecA-2 domain is flexible, and it is possible that the CTD closing-in and moving-out motions into the RNA-binding cleft contribute to the duplex-unwinding after each ATP hydrolysis cycle. Our BACTH and molecular docking studies showed that NS4b interacts with the CTD of the helicase. We performed molecular dynamics simulations on the NS4b-docked structure of NS3 to see if this interaction leads to any motion in the CTD, as we interpreted from our intrinsic tryptophan fluorescence studies. An overlay of the end of MD simulation trajectory structure with that of the initial structure (input of MD production run) is shown in Figure 4D. As can be seen from the overlay of NS3 structures (NS4b is not shown in the figure for clarity), the RecA and the protease domains did not show significant motion. There was no significant movement of the center of mass of these domains during the entire trajectory. However, a center-of-mass analysis on the CTD showed that, compared to the initial structure (t = 0 ns in Figure 4D), the t = 20 ns structure moved by 3.3 Å, away from the RecA domains. A RMSD and RMSF analysis of the trajectory also showed major deviation in residues corresponding to the CTD within the time-scale of the simulation. These results also suggest that CTD domain motions, probably on the hinge connecting the RecA-2 domain to the CTD domain, away from and towards the RecA domains are possible. The CTD domain motions may contribute to the RNA duplex unwinding activity of the helicase, as proposed earlier [33,46]. Furthermore, as revealed by our MD simulation results, the interaction of NS4b with the helicase CTD brings about similar motion in the CTD.

To provide more direct evidence for the CTD domain motion into the RNA-binding cleft, upon NS3–NS4b interaction, we attempted to crystallize NS4bN57 with NS3Hel. However, despite several crystallization attempts, we were not successful in co-crystallizing NS4bN57 along with NS3Hel. Crystals that we obtained had only NS3Hel and did not contain NS4bN57 (data not shown). Attempts to express a smaller fragment of NS4b N-terminus (residues 1–36) or an N-terminus disordered region along with scaffolding proteins are ongoing in the lab. We also attempted to introduce a FRET pair of fluorophores between the CTD and RecA-2 domain to study the domain motion in the helicase by single-molecule FRET experiments. Unfortunately, the introduction of cysteine or the introduction of an unnatural amino acid that is a photoactivable cross-linker in the CTD lead to protein misfolding (very low solubility). Attempts to express CTD alone were unsuccessful. All of these observations suggest that CTD plays an important role in the structure and function of the helicase, and mutations in the subdomain may not be tolerated.

### 3.5. NS3 Helicase Activity on the 3′-SL of the 3′-UTR Is Enhanced upon NS4b Interaction

Similar to many other SF2 helicases, flavivirus helicases show very poor in vitro helicase activity until an interacting protein enhances their activity. For dengue virus NS3 helicase, interaction with NS4b is shown to enhance its in vitro helicase activity [28]. Does the interaction between NS4b and NS3 helicase CTD cause the helicase activity enhancement? To test this, we optimized a helicase assay for NS3 in the presence of NS4b, using either an RNA or a DNA oligo corresponding to the 3′-SL region of the DENV serotype 1 sequence. Secondary structure predictions on the sequences show that both RNA and the DNA oligos can form a stable stem-loop structure (−29.00 and −15.83 kCal/mole of folding energy, respectively, as per RNAstructure software prediction) that mimics the natural substrate of the helicase (Figure 5A).

We then purified NS4b without using LDAO detergent in the protocol. We purified the protein without the detergent for the following reasons: (i) the LDAO detergent present in the NS4b preparation (purified following LDAO detergent extraction protocol) may interfere with NS3 helicase activity, (ii) our studies showed that NS4b membrane insertion is not necessary for interaction with NS3, and (iii) the dynamic oligomerization of NS4b is observed even in the presence of detergent micelles.

NS4b purified without using detergent extraction showed far-UV CD spectra that almost overlapped with that of the protein purified in the presence of LDAO (Figure 5B) and formed large soluble oligomers (Figure 5C). However, we did not see a monomer or dimer population. This implies that there are no significant differences in the overall folding of NS4b with or without the membrane environment. We also expressed and purified the N-terminal 57 residues region of NS4b (NS4bN57). Surprisingly, NS4bN57 also oligomerizes, eluted at a volume corresponding to ~340 kDa (monomer mass is 7.5 kDa) on a Superdex-200 column. Gel-filtration chromatography on a mixture of NS4bN57 and NS3 (1:1 molar ratio) showed that the proteins formed a stable complex. NS3 coeluted along with NS4bN57 oligomer (Figure 5D and inset in the right panel). This observation further validates our BACTH analysis and docking prediction that only N-terminal 57 residues of NS4b is enough to interact with NS3. We used either the full-length NS4b or the NS4bN57 proteins to test their effect on the helicase activity of NS3Hel.

Then, using the 3′-SL forming RNA and DNA oligos as substrates and purified NS3 and NS4b proteins, we optimized the NS3 helicase enzyme assay, adapting a gel mobility shift assay described earlier [20]. The rationale of the assay is that the stem-loop structure is more retarded in the electrophoretic gel compared to the resolved (single-stranded) oligo. DENV NS3 helicase can use duplex DNA or duplex RNA as a substrate in vitro. In a control experiment, we analyzed a sample of the heat-denatured DNA oligo (without allowing annealing) next to a sample of the annealed 3′-SL oligo in a gel-mobility shift assay. The annealed oligo is retarded more (ran above to the denatured oligo) than the denatured (completely unwound) oligo (Figure 5E), confirming that the 3′-SL oligos that we used formed stem-loop structures as predicted. Adding NS3Hel protein alone to the 3′-SL oligo did not show a band corresponding to unwound (Figure 5E, Lane 2), signifying that are no unknown components in the NS3 helicase protein preparation that may cause the unwinding of the stem-loop. Figure 5F shows a representative gel image from the helicase stem-loop unwinding experiment using the DNA oligo as substrate. In the presence of NS3 and ATP, a significant lower band corresponding to the unwound 3′-SL DNA oligo is seen (marked ssDNA in Figure 5F). A band corresponding to the resolved DNA (ssDNA) is seen only when NS3 was present in the assay mixture. No ssDNA band was seen when only NS4b was added to the assay or when ATP was omitted in the assay. This confirmed that the DNA oligo that we designed did form a stable stem-loop structure as predicted, and the DNA duplex unwinding activity observed is specifically because of the NS3Hel helicase activity. Two hours after the assay start time (time of addition of ATP to the reaction mixture), most of the stem-loop band intensity is reduced, and the intensity of the band corresponding to the resolved DNA increased. For a comparison of the helicase activity of NS3Hel in presence of NS4b or NS4bN57, we chose the one-hour time point as the end point of the assay, as this allowed us to assess the increase in helicase activity better.

We also performed the NS3 helicase assay using 3′-SL RNA oligo as substrate—the natural substrate of the helicase. A representative image of the gel from the gel mobility shift assay is shown in Figure 5G. Unlike the 3′-SL DNA oligo, the RNA oligo showed few lower bands in addition to the band corresponding to the stem-loop. However, when the oligo was incubated with NS3Hel and ATP in the assay buffer, the band intensity corresponding to the unwound oligo (marked ssRNA in Figure 5G) increased, with a concomitant decrease in the intensity of the stem-loop band. This confirms that the NS3Hel can use the 3′-SL RNA oligo as a substrate and can unwind it. Our observation is the first in vitro biochemical evidence that DENV NS3 helicase uses 3′-SL as a substrate. However, we noticed that when NS4b or NS4bN57 proteins were added to the assay mixture along with the RNA oligo, a major proportion of the oligo precipitated and did not enter the gel (data not shown). This probably happened because the hydrophobic dye (FAM) that we conjugated to the oligo for detection after the electrophoretic run binds non-specifically to the hydrophobic patches on NS4b. As a result, we could not use the RNA oligo in the helicase assays with NS3–NS4b or NS3–NS4bN57 mixtures.

We also used a fluorophore and a quencher attached single-stranded RNA oligo in a molecular beacon assay to test DENV NS3 helicase assay adapting a protocol described earlier [42]. As schematically depicted in Figure 5H, the fluorophore–quencher pair attached to an RNA oligo is annealed along with a complimentary un-labeled oligo. If NS3Hel unwinds the duplex RNA, the labeled oligo can form a stem-loop structure, thus bringing the fluorophore and the quencher into FRET distance. The extent of quenching is used as a measure of NS3 helicase activity (one representative CY5 fluorescence intensity vs. time of the assay is shown in Figure 5H, next to the schematic). The sequence of the labeled-RNA oligo is not that of the 3′-SL region though. We used this assay to measure the helicase activity on the RNA duplex in the presence of NS4b and NS4bN57 in place of the gel-mobility shift assay. In the presence of NS4b or NS4bN57, the helicase activity on the RNA duplex was enhanced (table in Figure 5H). However, we noticed that the fluorescence signal was very noisy when NS4b was included in the assay, resulting in large standard errors in the experiment. Since we did not notice any difference between the 3′-SL RNA and DNA oligo as the substrate of NS3Hel, we used the 3′-SL DNA oligo as a substrate for further studies to quantify the helicase rate enhancement in the presence of NS4b or NS4bN57.

To assess the affect of NS4b on NS3 helicase activity, we performed the helicase assay in the presence of different molar ratios in excess of NS4b to NS3Hel using 3′-SL DNA oligo as substrate. A representative gel image from the helicase assays is shown in Figure 6A. The ratio of background-corrected pixel intensities of the unwound oligo band (ssDNA) to the sum of pixel intensities from the stem-loop band and unwound (ssDNA) band is expressed as % helicase activity (refer to the Section 2 for details). The activity of NS3Hel protein at the assay end point (1 h post addition of ATP into reaction mixture) is taken as 100% activity. The pixel intensity of the unwound (ssDNA) band at the zero time point (stopping the reaction within five minutes of adding ATP to the assay mixture), if any, is subtracted in each case to normalize the zero time point activity to 0%. In the presence of a three-molar excess of NS4b, NS3Hel helicase activity increased by 53 (±16)% over and above the NS3Hel-alone activity (Figure 6B). Similarly, in the presence of NS4bN57, the activity increased by 22 (±14)% (Figure 6C). Notably, though NS4bN57 showed interaction with NS3Hel at 1:1 and 1:2 molar ratios, the interaction did not result in an increase in the helicase activity of NS3Hel to the same extent as what full-length NS4b could bring about. It is possible that, though the N-terminus 57 residues region of NS4b is enough to interact with NS3 helicase, other regions of the NS4b may be important for helicase activity modulation, in ways that we cannot explain from our study.

We also optimized an assay for the ATPase activity of the helicase and used it to determine the Michaelis–Menten kinetic constants of the reaction. The reaction initial velocities at different substrate concentrations fit an MM-kinetics model using a non-linear regression method as shown in Figure 6D. In the presence of NS4b (at a 1:3 NS3 to NS4b molar ratio), the V_max_ of the reaction increased to 0.560 ± 0.051 µmoles/min from 0.092 ± 0.018 µmoles/min for the NS3Hel-alone reaction, implying a nearly 6-fold catalytic rate enhancement of the helicase upon NS4b interaction. Assuming that the ATPase activity and RNA duplex unwinding activity are correlated, an increase in RNA unwinding activity of the helicase to the same extent (as ATPase activity) is expected.

Taken together, our results from gel-shift mobility assays using RNA and DNA oligos predicted to form a 3′-SL structure, molecular beacon assays with RNA oligos, and MM-kinetic studies show that the 3′-SL in the 3′-UTR region of the DENV genome is a cognate substrate for NS3 helicase and the interaction of NS4b with NS3 helicase enhances the in vitro helicase activity. These results also validate our observations from BACTH assays and molecular docking studies that the N-terminus 57 residues region of NS4b is enough to interact with NS3 helicase.

## 4. Discussion

Flavivirus non-structural proteins interact with each other and with host proteins in a complex and dynamic manner [14,48]. Several studies established the criticality of NS3–NS4b interaction [18,24,29] in the DENV replication. Furthermore, targeting the NS4b–NS3 interaction in DENV and other flaviviruses has been proposed as a therapeutic strategy. Recently, a small-molecule drug, JNJ-A07 (a hit from a large-scale cell-based anti-DENV 2 screen), has been shown to potently inhibit viral replication by targeting the NS4b–NS3 interaction [35]. However, the interaction between NS3 and NS4b, and its significance to the enzymatic functions of NS3, is not completely understood. With the objective of characterizing the interaction between NS3–NS4b and providing a structural and mechanistic insight into the interaction, we characterized the DENV 1 NS3–NS4b interface and explained a possible structural mechanism for the helicase activity enhancement of NS3 upon the interaction.

Earlier studies by Umareddy et al. (2006) [28] showed, using a yeast-two-hybrid assay and a pull-down assay on DENV-infected cell lysates, full-length NS4b interacts specifically with the NS3 helicase subdomains RecA2 and CTD. Our BACTH results and docking simulations are consistent with their results. However, in their assays the N-terminal half (1–135) or the C-terminal half of NS4b (136–248) alone did not interact with NS3. Interestingly, NS3 also interacts with NS5 through the RecA-2 and CTD subdomains [25,49]. It is possible that both NS5 and NS4b interact with NS3 subdomains two and three and form a tri-partite complex during negative-strand synthesis [48].

There are a few other studies wherein the NS3–NS4b interaction interface is mapped. Zou et al. (2015) [24] using purified proteins of NS3 full-length or different subdomains of the helicase and NS4b showed that the RecA2 and CTD of NS3 participate in interaction with NS4b. Our docking studies corroborate their findings. The binding strength of the interaction reported in their study (K_D_ of 222 nM for DENV2 and 530 nM for DENV4 NS3Hel–NS4b) are comparable to what we observed in our BLI studies (K_D_ of 508 nM). They also tested interaction between a peptide with a sequence corresponding to a loop connecting the NS4b TM3 and TM4 with NS3. Their rationale for selecting this region is based on the predicted membrane topology of NS4b, wherein the TM3–TM4 loop is the only region that would face the cytosolic side of the RO, wherein NS3 is localized. Interestingly, the K_D_ values estimated for NS3–NS4b cytosolic loop binding (1.6 μM) is very high compared to those observed with full-length NS4b and helicase. This implies that there may be other regions of NS4b that can interact with NS3. Further, using NMR studies with the cytosolic loop peptide and NS3 helicase domain, followed by genetic analysis, they fine-mapped the interacting residues to Q134 and G140. In another independent study, Chatel-Chaix et al. (2015) [18] made an identical finding, that the Q134 residue of the TM3–TM4 cytosolic loop is the determinant of NS4b interaction with NS3. Interestingly, in their studies, alanine substitutions at DENV2 NS4b residue positions 28 (L28A) and 87 (M87A) also showed the apparent loss of interaction with NS3, albeit to a much lesser extent than the cytosolic loop mutations (Q134A, G140A, and M142A).

The studies described above on the NS4b–NS3 interaction interface mapping specifically looked for interaction between the NS4b TM2–TM3 cytosolic loop residues and NS3. Their rationale being, as per the predicted membrane topology of the NS4b [29], the 2K signal preceding NS4b would lead the protein into the membrane from the ER lumen side, and thus, the N-terminal 100 residues region is placed in the ER lumen. In that scenario, the cytoplasmic loop between TM3–TM4 is the only accessible region on the cytosolic side of the RO for NS3 interaction. We have a differing view about the localization of N-terminal 100 residues of NS4b. Our proposition is that the N-terminal 100 residues are disordered and extend away from the membrane surface into the cytosol side of the RO (as depicted in Figure 1A, schematic of the RO on ER). This would allow its interaction with NS3 and other interacting partners. The reasons for our proposal are as follows: (i) Miller et al. (2006) [29], based on TM helix predictions and biochemical studies, proposed a membrane topology for NS4b, which the above-mentioned NS3–NS4b interaction studies were based on. However, the proposed topology model does not exclude the possibility of the N-terminal region flipping to the cytosol side of the RC after proteolytic processing at the C-terminal end of the 2K signal peptide. This may happen in a way similar to what is proposed for the TM5 of NS4b—TM5 flips from the cytosolic side to the ER lumen side after the NS4b–NS5 boundary is cleaved [18]. Alternatively, after the cleavage at the 2K signal peptide end, the N-terminus of NS4b may come out of the membrane to the cytosolic side, thus flipping the whole NS4b orientation in the membrane opposite to what Miller et al. [29] proposed, as depicted in our membrane topology diagram for NS4b in Figure 1B; (ii) Zou et al. (2014) [19], based on their fluorescence protease protection assay results with DENV 2K-NS4b(1-93)-EGFP, inferred that the N-terminus 100 residues may position on the ER lumen side, as well as on the cytosol side, of the RO; (iii) Zou et al. (2015) [24] could immuno-precipitate NS4b from DENV-infected cell lysates without the detergent extraction of the protein. Since the majority of the NS4b, except for the N-terminal 100 amino acids, insert into the membrane, the only explanation for this result is that the N-terminus is on the cytosolic side of the RC, where NS3 is also present; (iv) The N-terminal 95 residues of NS4b are required for the modulation of IFNα/β signaling [31] in DENV-infected cells, probably through its interaction with the STAT-1 protein, a cytosolic protein. Thus, our proposal that the N-terminus region of NS4b extends like a tail into the cytosol for interaction with various protein factors is not unfounded. Consistent with this idea, recently Lu et al. (2021) [30] showed that the N-terminal 51–83 residues of NS4b are enough to interact with NS3 and enhance NS3 helicase activity. However, their study was done with peptides corresponding to different regions of the N-terminal region or the cytosolic loop connecting the TM3 and TM4 of NS4b (as Sumo-fusion proteins).

In our studies to map the interaction between NS3 and NS4b, we did not focus on any region of NS4b and NS3, unlike the studies mentioned in the above paragraphs. Our molecular docking simulations were performed with full-length proteins (including the protease domain of NS3) without a prior notion about the interaction. Our in vitro studies with purified proteins (full-length NS4b or NS4N57 and NS3Hel) to characterize the interactions validated our docking studies. Collectively, based on all of our studies to characterize the interaction, we conclude that the N-terminal, likely disordered, region of NS4b is the major determinant for interaction with NS3, supporting the findings by Lu et al. (2021) [30]. Based on these observations, we propose a mechanistic model for the NS3–NS4b interaction (Figure 7). Though speculative at this juncture, it is reasonable to think that the N-terminal disordered region of NS4b flips from the ER lumen side to the cytosol side (by a mechanism that cannot be explained from our studies) after the C-terminal end of the 2K signal is proteolytically processed. Then, the N-terminal end extends as a tail down from the membrane surface, runs lateral to the RecA-2 domain, and partially wraps around the CTD from the side opposite to the RNA binding cleft. However, whether only the N-terminus of NS4b flips to the cytosolic side, keeping the transmembrane helices in the orientation as proposed by Miller et al. [29] needs to be tested. Only in this scenario, both the N-terminus region of NS4b and the TM3–TM4 loop can interact with NS3 helicase simultaneously. As we noted in our docking analysis, a salt bridge between the K_10_ of NS4b and NS3 CTD E_568_ may stabilize the interaction to hold the NS4b N-terminal region around the CTD tightly. It would be interesting to see if mutating the residues, K_10_ from NS4b or E_568_ in NS3, would break the interaction between NS4b–NS3. We could not test this idea, as when we introduced mutations in the neighboring residues to E_568_ in NS3 (for using them in crosslinking studies), the mutant proteins did not express well, possibly because of the misfolding of the protein. Thus, this new interaction interface between NS4b and NS3 that we propose here, along with interaction with the loop regions between the TM helices proposed earlier, is likely to be the complete interaction interface between the two proteins. This complete map of the NS3–NS4b interaction interface should be considered in designing therapeutic strategies to block the interaction.

What does interaction with NS4b signify to the enzymatic functions of NS3? Since only the helicase subdomains are involved in interaction with NS4b, it is reasonable to assume that NS4b can influence the helicase activity of the protein. Consistent with this idea, Umareddy et al. (2006) [28] showed that NS4b could enhance the duplex RNA unwinding activity of NS3 by nearly two-fold at a 1:2 NS3 to NS4b ratio. However, a mechanism that can explain the rate enhancement is not known. In our study, we used an RNA or a DNA stem-loop that would be analogous to the natural substrate of NS3—the 3′- stem-loop in the 3′-UTR. This stem-loop (3′-SL) is likely to be the first substrate for NS3 during negative-strand synthesis. Our biochemical characterization showed that the 3′-SL is indeed a cognate substrate for the NS3 helicase. It is important to note that earlier studies with NS3 helicase used either an RNA duplex or stem-loop forming sequences rather than 3′- SL from the 3′UTR of the viral genome.

We also studied conformational changes in NS3 and NS4b proteins upon interaction. Though not conclusive, our intrinsic tryptophan fluorescence and MD-simulation studies suggested that NS4b–NS3 interaction may increase the dynamic motion of the CTD (subdomain 3) of NS3. Based on the crystal structures of the DENV NS3 helicase in different ligand-bound (an ATP-analogue, Mn^2+^, intermediates of ATP hydrolysis, and a ssRNA fragment) states, Luo et al. (2008) [23] proposed a mechanism of NS3 nucleoside triphosphatase activity and the coupled dsRNA unwinding activity. As revealed in these crystal structures, ssRNA is held in the RNA-binding cleft by interactions with several residues in all three subdomains, although the majority of the interactions are with RecA-1 and -2 subdomain residues (primarily with RNA backbone phosphoryl oxygen). They also noted that, in the ssRNA-bound structure, subdomain 3 (CTD) rotates by 11 degrees, away from the RecA domains, to make the RNA-binding cleft wider. Based on these observations, they proposed that subdomain motions after dsRNA binding in the RNA-binding cleft (CTD opening and closing onto Rec-A subdomains) may provide the wrenching force to push the duplex through the ‘separation pin’—a β-hairpin structure capping the RNA-binding cleft of NS3. A similar proposal—dynamic CTD (subdomain 3) motions during the unwinding action of helicase to close the RNA-binding cleft—was made for Zika virus NS3 helicase [33,46] and hepatitis C virus NS3 helicase [50]. In accordance with this hypothesis, a molecular dynamics simulations study on the dengue virus NS3 helicase structure in complex with RNA and ATP hydrolysis intermediates predict dynamic conformational changes in the protein, explaining the allosteric mechanism connecting the helicase activity to the ATP hydrolysis [47]. Interestingly, in the NS4b-docked NS3 structure (this study, Figure 2C), the disordered region of NS4b is closely apposed to the ‘Rec A2-CTD hinge’ of NS3. Furthermore, our MD simulation studies on the NS3–NS4b docked structure are also indicative of CTD domain motions. Our study provides experimental evidence, though not direct evidence, in support of this proposed mechanism. Tertiary structure changes in the NS3–NS4b complex that we observed correlate well with the rate of enhancement of the helicase activity. At higher NS3 to NS4b molar ratios, wherein helicase activity is higher, the intrinsic tryptophan intensity significantly enhanced. This change can be interpreted as more rapid dynamic motions in the subdomains resulting in enhanced helicase activity.

Could the NS4b N-terminus interaction with NS3 RecA2 and CTD domains enhance the subdomains motion to increase the helicase activity? Our results of our molecular docking, and far-UV CD and intrinsic tryptophan fluorescence studies and MD simulations, as well as helicase assays, are indicative of such a mechanism. As described in the Section 3 (refer to Figure 2C and Figure 4B), it is reasonable to interpret the observed change in intrinsic tryptophan fluorescence in the NS3–NS4b complex (but no far-UV CD changes) as primarily due to CTD moving closer to the RecA domains. Taken together, a plausible explanation is that NS4b interaction with NS3 helicase increases the RNA duplex unwinding activity by increasing the CTD motions (towards and away from the RecA domains for each cycle of unwinding/RNA translocation).

## 5. Conclusions

We propose a mechanism of the concerted action of NS3-interacting protein complexes in viral negative-strand synthesis. As depicted in Figure 7, the N-terminal disordered region of NS4b translocates into the cytosolic side of the RO, after the proteolytic processing of the 2K C-terminus signal. NS3 interacts with the NS4b N-terminus region, and this would tether it to the membrane. The NS4b N-terminus region wraps around the RecA-2 and CT subdomains of the helicase and modulates the subdomain motions during the dsRNA unwinding step. NS3 also interacts with NS5 through the CTD. Through this interaction, the NS5 is also positioned on the 3′-end of the genome, where negative-strand synthesis starts from. As a result of these interactions, a tripartite complex of NS3–NS4b–NS5 may form in the RC, where the NS3–NS4b complex will precede NS5. The conserved stem-loop of the 3′-SL (in 3′-UTR domain III) is then unwound by NS3 helicase. NS4b modulates the helicase activity by controlling the motion of the CTD and thus modulating the duplex RNA binding cleft conformation.

The interaction of NS3 with NS4b and NS5, being critical for flavivirus replication, can potentially be targeted in therapeutic development against flaviviruses. With a complete map of the NS3–NS4b interaction and understanding the mechanism of NS3 helicase activity modulation by the interaction will greatly help in the better design of the inhibitory drugs.

## Figures and Tables

**Figure 2 viruses-14-01712-f002:**
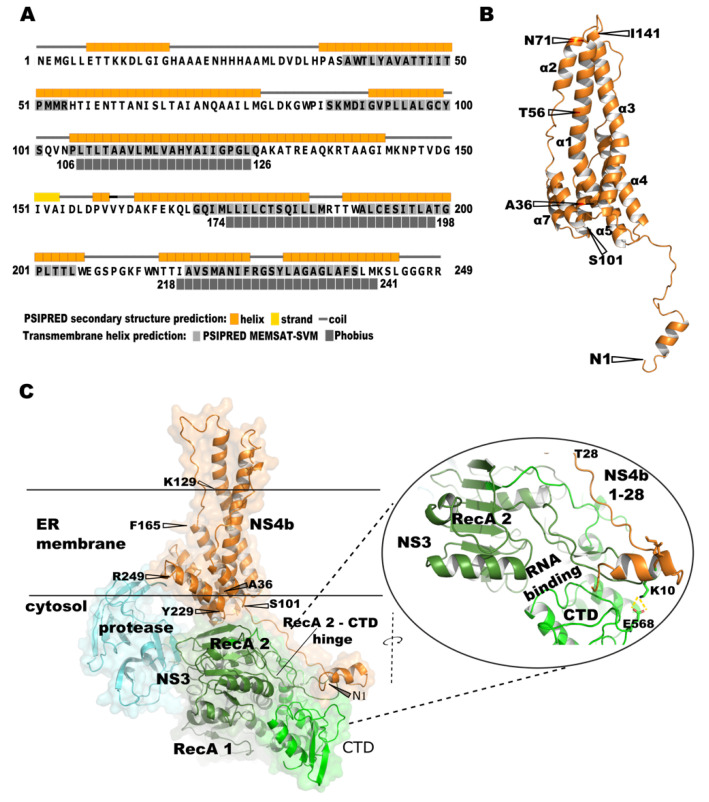
Membrane helix predictions, *ab-initio* structure prediction for NS4b and the molecular docking pose of NS4b on NS3. (**A**) DENV serotype 1 NS4b sequence along with PSI-PRED secondary structure prediction (orange boxes for helix and black solid lines for coil) is shown. Transmembrane helix predictions by PSIPRED MEMSAT-SVM (light-gray-boxed regions) and Phobius (dark gray boxes below the sequence) are also shown. (**B**) The AlphaFold 2.0-predicted structure model of DENV serotype 1 NS4b. The seven helices predicted are labeled (a1–7). Residue positions bounding each helix are labeled. The N-terminal residues N_1_ to A_36_, disordered region, are marked. Position of residue 56 (T56) in the sequence is also marked; (**C**) Best-docked pose of NS3–NS4b from HADDOCK molecular docking. The NS4b (orange cartoon and surface representation) and NS3 (different shades of green cartoon representation) docked pose are shown in an orientation likely to be seen in the RO. The likely membrane boundary positions are shown with black lines. The boundaries of TM3 (as per Miller et al. (2006), [29]), S_101_ and K_129_, and TM4 (F_165_) are marked. The protease subdomain of NS3 is colored in cyan. The RecA 2–CTD hinge in the NS3 helicase is marked. The zoomed-in section shows the salt bridging residues K_10_ (from NS4b) and E_568_ (from NS3) in stick representation. Structure figures are prepared in PyMol.

**Figure 3 viruses-14-01712-f003:**
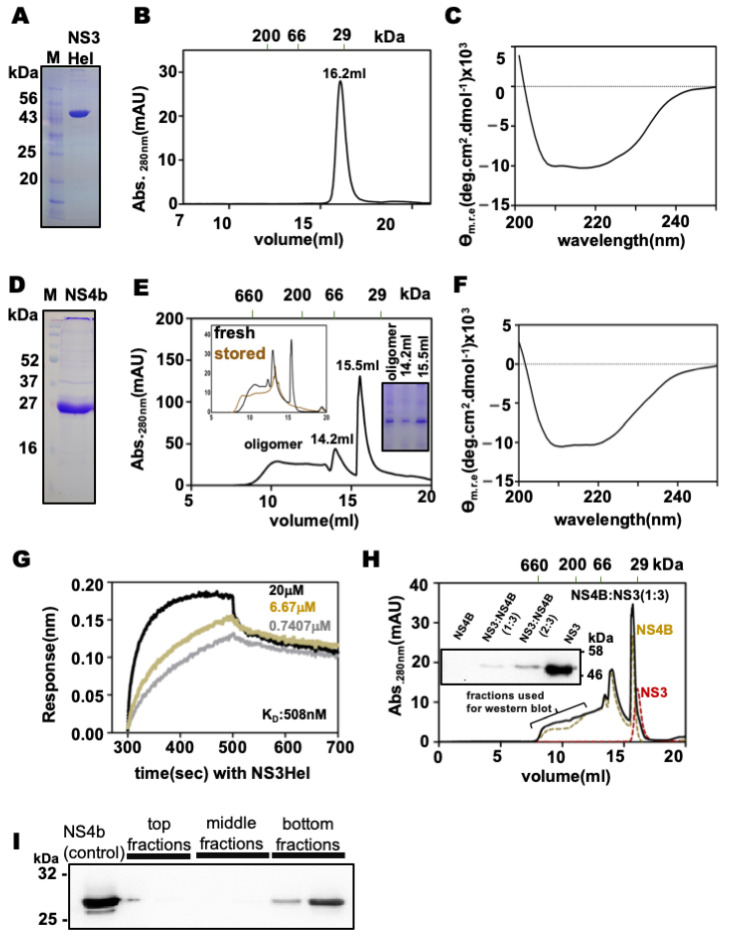
Biochemical characterization of NS3, NS4b-purified proteins, and their interaction. (**A**) SDS-PAGE of the purified NS3Hel protein. M is the protein molecular-weight marker lane. Staining was done with Coomassie Brilliant blue R250 following standard protocols. (**B**) Gel-filtration chromatograph of NS3Hel protein. The protein was loaded (0.2 mg) on a Superdex-200 column and eluted with TN buffer ata 0.5 mL/min flow rate. Elution positions of molecular-weight marker proteins are marked with the corresponding molecular weight. (**C**) Far-UV CD spectrum recorded on a purified NS3Hel protein. (**D**) A representative gel image from a SDS-PAGE analysis of recombinant NS4b protein purity. M is the protein molecular-weight marker lane. (**E**) Gel-filtration chromatograph of NS4b. The protein was loaded (1.5 mg) on a Superdex-200 column and eluted with TN buffer with 0.05% LDAO at a 0.5 mL/min flow rate. Inset in the right: SDS-PAGE analysis on fractions from different peaks in the chromatograph. Inset in the top left: overlay of chromatographs for fresh NS4b preparation (black line) and for the protein sample two hours after storage at 4 °C. (**F**) Far-UV CD spectrum recorded on the purified NS4b protein. (**G**) NS3–NS4b binding studied by BLI: sensograms (response curves) recorded with the indicated concentrations of the NS3Hel protein with NS4b bound to the sensor. (**H**) Gel-filtration experiments to study NS3–NS4b interaction: a mixture of NS3Hel and NS4b proteins in a 1:3 molar ratio (assuming monomer molecular masses) is loaded onto a Superdex-200 column and eluted in TN buffer. Overlay of the NS3Hel alone (red dotted line) and NS4b alone (gold dotted line), along with a chromatograph of NS3Hel:NS4b (black solid line), is shown. Western immuno-blotting on the fractions marked (in the NS4b oligomer elution region) was done with anti-NS3 rabbit polyclonal antibody. (**I**) Western immuno-blotting analysis of fractions from LUV:NS4b co-floatation assays: the lane marked NS4b (control) is NS4b loaded in the gel as a positive control for anti-penta His antibody-HRP conjugate reactivity. Lanes marked top, middle, and bottom have aliquots from fractions (1 mL each) collected from the top to bottom of the sucrose gradient after the ultracentrifugation run.

**Figure 4 viruses-14-01712-f004:**
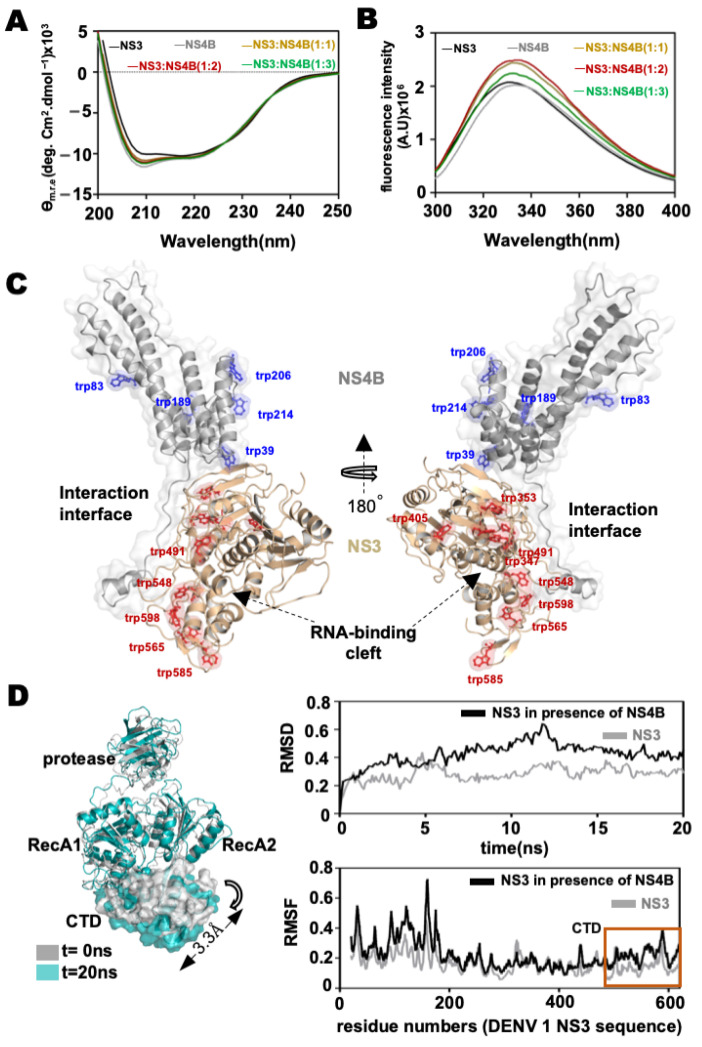
Conformational changes in NS3 upon NS4b interaction (**A**) Overlay of far-UV CD spectra recorded on NS3Hel, NS4b, and a mixture of the proteins in different molar ratios. (**B**) Overlay of the intrinsic tryptophan fluorescence spectra of NS3Hel, NS4b, and a mixture of the proteins in different molar ratios. (**C**) Cartoon and transparent surface representation of NS3–NS4b docked pose (from the molecular docking experiments). Tryptophan sidechains in both proteins are shown in stick representation. Tryptophan positions in NS4b are labeled in blue and for NS3 in a red-color font. (**D**) Left: overlay of the NS3 structure (MD simulation input structure; t = 0 ns) and the final structure of the MD simulation trajectory (t = 20 ns, cyan cartoon and surface representation). The movement of the center of mass (COM) of the CTD in the MD simulation run is shown by a curved arrow, and the distance between the COMs (between t = 0 and t = 20 ns structures) is shown. Right: plots from root mean square deviation (RMSD in nm) and root mean square fluctuation (RMSF in nm) analysis of the trajectory. The orange box in the RMSF plot denotes the CTD region of the NS3 helicase domain sequence.

**Figure 5 viruses-14-01712-f005:**
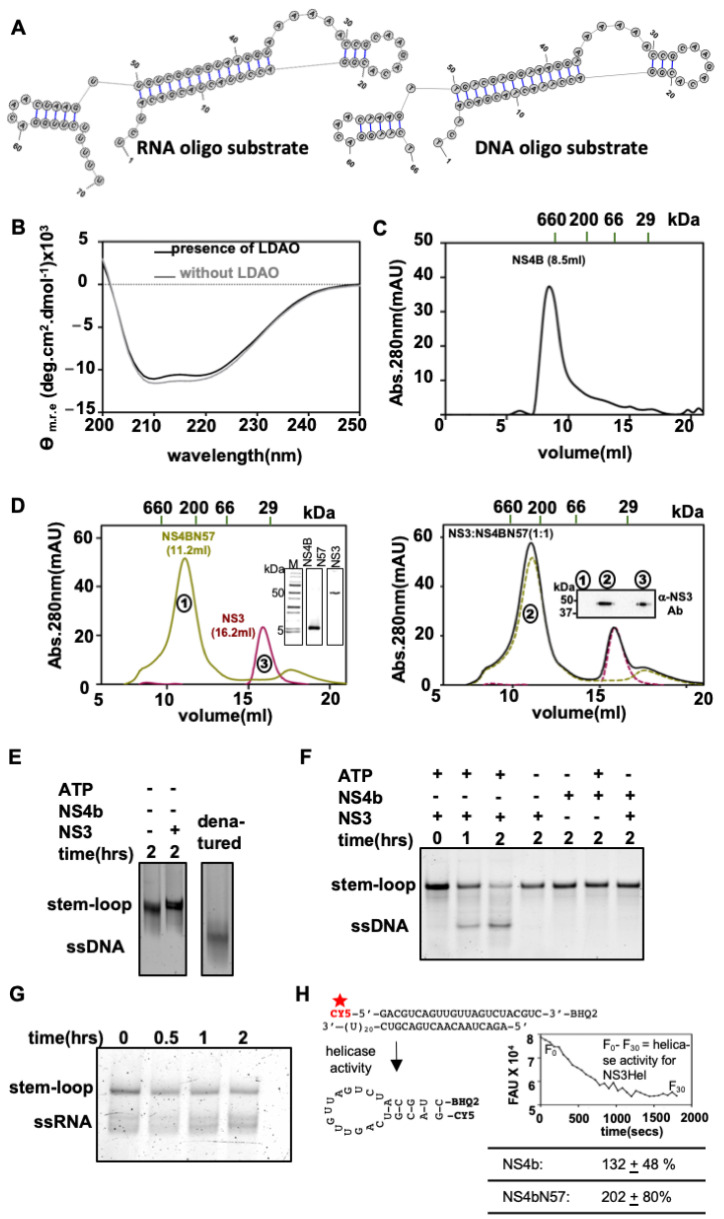
NS3 helicase assays with an RNA and a DNA stem-loop forming oligo substrates. Schemes follow another format. If there are multiple panels, they should be listed as: (**A**) predicted structures for the RNA and DNA oligos that are used as substrates of NS3 helicase in the gel mobility shift assays. (**B**) Overlay of far-UV CD spectra recorded for NS4b, either with protein purified using LDAO (black line) or without the detergent (gray line). (**C**) Gel-filtration chromatograph showing the elution of the NS4b protein purified without detergent extraction on a Superdex-200 column. (**D**) NS3–NS4bN57 interaction analyzed by gel-filtration chromatography. Left: overlay of NS3Hel and NS4bN57 elution profiles on a Superdex-200 column. The inset shows a gel image with fractions from the elution peak in each case analyzed on SDS-PAGE. Right: overlay of gel-filtration chromatographs for NS3Hel, NS4bN57, and a mixture of NS4bN57:NS3Hel at a 1:1 molar ratio. The inset shows Western immunoblotting done on elution fractions from each peak in the chromatographs (marked 1, 2, and 3 numbers in circles) for the presence of NS3 in those fractions. (**E**) Electrophoretic mobility shift assay on 3′-SL DNA oligo. Lane 1: the DNA oligo incubated for 2 h in assay buffer alone, Lane 2: the DNA oligo incubated with the NS3 protein in the assay buffer, and Lane 3: the oligo is heated to 95 °C for 5 min to denature (unwind the stem-loop) and loaded onto the gel without allowing it to re-anneal. (**F**) Electrophoretic mobility shift assays for NS3 helicase activity. Gel mobility shift helicase assays with DNA stem-loop oligo. Lane 1: the DNA oligo mixed with NS3Hel and ATP and immediately the reaction was stopped by adding the EMSA loading dye (0 h time point). Lanes 2 and 3: same as in Lane 1, but the reaction was stopped one hour and two hours after the addition of ATP, respectively. Lane 4: no ATP was added to the reaction (NS3Hel-alone and control). Lane 5: the oligo was incubated with only NS4b protein for two hours. Lane 6: only NS4b and ATP were added to the reaction (no NS3Hel control). Lane 7: only NS4b and NS3Hel are added to the reaction (no ATP control). After the electrophoretic run, gels were imaged, and the pixel density of bands corresponding to ssDNA and the stem-loop were quantified using ImageJ software. The band corresponding to the resolved stem-loop DNA oligo is marked as ssDNA. (**G**) Electrophoretic mobility shift assay for NS3 helicase activity on RNA 3′- stem-loop oligo. Lane 1: 0 h. time point of the assay. Lanes 2, 3, and 4 have 3′-SL oligo incubated with NS3Hel in the assay buffer for 0.5, 1, and 2 h, respectively. (**H**) Molecular beacon assay of NS3 helicase. Sequences of the labeled RNA oligos used in the assay are shown. A red star mark (above ‘CY5’) is to schematically represent that the fluorescence is not quenched when both oligos form a duplex as depicted. The likely stem-loop structure of the labeled-oligo formed after duplex unwinding by the helicase activity is shown below. A representative (from one of the experimental replicates) CY5-fluorescence time scan is shown to the right (above), and helicase rate enhancements, estimated from the fluorescence quenching data, in the presence of NS4b or NS4bN57 are shown in the table below the fluorescence time scan.

**Figure 6 viruses-14-01712-f006:**
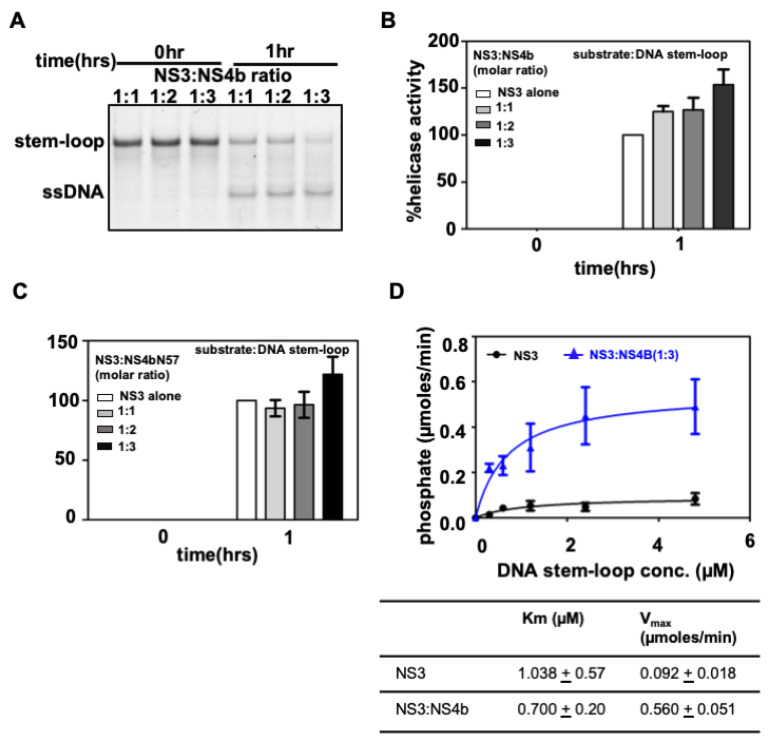
Quantification of NS3 helicase activity rate enhancement in the presence of NS4b. (**A**) EMSA on the DNA stem-loop substrate in the presence of different NS3 to NS4b molar ratios. A representative gel image from the experimental triplicates is shown. (**B**) Bar diagram showing mean (three replicates) helicase activity increase (on the DNA stem-loop) over and above the NS3Hel alone activity in the presence of NS4b. The percent helicase activity is estimated by taking the ratio of resolved DNA band (ssDNA) intensity over the sum of resolved and stem-loop band intensities and expressing it as a percentage. Helicase activity estimated with NS3Hel-alone 1 h after the start of the assay is taken as 100%. The 0 h time point activity estimate is normalized to zero in each case. Standard errors in each experiment are estimated and shown as a capped line, along with the mean values (height of each bar). (**C**) Bar diagram showing mean (three replicates) helicase activity increase over and above the NS3Hel-alone activity in the presence of NS4bN57. (**D**) Michaelis–Menten kinetics curve fitting of the NS3Hel activity. The initial velocities at different indicated substrate (DNA stem-loop) concentrations are estimated from the linear ranges of kinetic traces of the phosphate release from the ATPase activity of the helicase. The kinetic data is fit to the MM-kinetics model by the non-linear regression method. The Michaelis–Menten kinetic constants estimates from the curve fit are shown in the table below.

**Figure 7 viruses-14-01712-f007:**
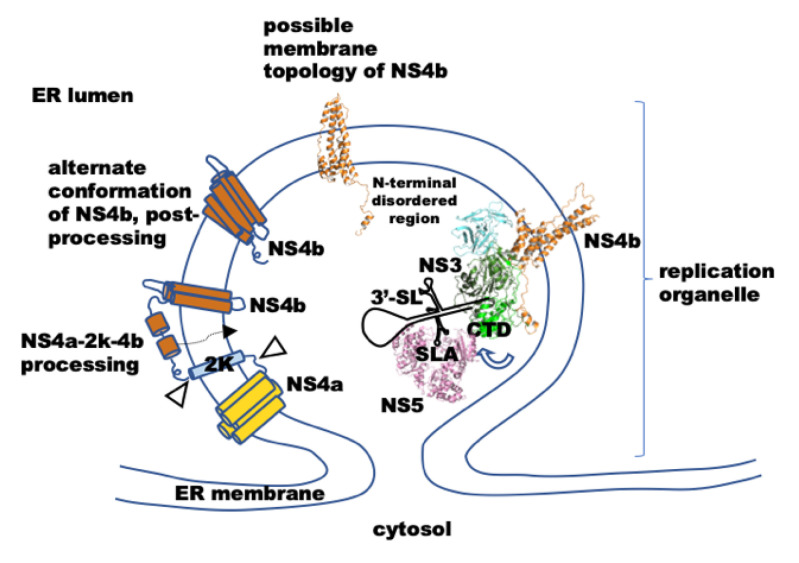
Proposed mechanism of the NS4b–NS3 interaction in the DENV replication organelle and the helicase activity modulation.

## Data Availability

Not applicable.

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
