# Peer review of "Dengue Virus NS4b N-Terminus Disordered Region Interacts with NS3 Helicase C-Terminal Subdomain to Enhance Helicase Activity"

_viruses, 2022, doi:10.3390/v14081712_

Round 1
Reviewer 1 Report
NS3 plays critical roles that are essential for viral replication and represents a very interesting target for the development of specific antiviral inhibitors. Characterization of the interaction between NS3 and NS4b and a possible mechanism of helicase activity modulation by NS4b are important issues. However, unclear how this novelty is better than Virus Res. 2021 Apr 2;295:198318., J Virol. 2015 Apr;89(7):3471-83. and J Virol. 2015 Jul;89(14):7170-86..
Author Response
As the reviewer pointed out, there were three different studies Zou et al. J Virol. 2015 89(7):3471-83 (reference 24 in our manuscript), Chatel-Chaix et al. J Virol. 2015 Jul;89(14):7170-86 (reference 18 in our manuscript) and Lu et al., Virus Res. 2021 Apr 2;295:198318 (reference 30 in our manuscript) which characterized the interaction between NS4b and NS3 of DENV.
As we explained in our manuscript Introduction and Discussion sections, (lines 113-136 in the Introduction section; and in the fourth paragraph of the Discussion section, starting from line 1001), studies by Zou et al. (2015) and Chatel-Chaix et al. (2015) specifically looked at the interaction between the TM3-TM4 loop of NS4b, which is proposed to be presenting to the cytosolic side of the replication organelle based on the membrane topology proposed by Miller et al., (2006) (reference 29 in our manuscript). On the contrary, Lu et al. (2021) reported that the region 51-83 residues in the N-terminal of NS4b, but not the cytosolic loop, is required for interaction with NS3 and the helicase activity enhancement. In this study as well, peptides corresponding to different regions of N-terminal region of NS4b along with a sumo-tag are used for studying the interaction strength and in the helicase assays.
In contrast to the above-mentioned studies, we took a different and novel approach for fine-mapping the interaction interface between DENV NS3 and NS4b. Several reports based on various biochemical studies (as we described in the Introduction section, lines 125-136) indicated that the N-terminus region of NS4b is on the cytosolic side of the RO (unlike the membrane topology model for NS4b adapted by the three studies mentioned above), where NS3 is also localized. Hence, we studied the interaction between NS3 and NS4b by bacterial two-hybrid assays without specifically focusing on any particular region of the protein and took an unbiased approach (neither biased to the cytosolic loop nor to the N-terminal region) to fine map the interaction. Our bacterial two-hybrid assay results unequivocally showed that the N-terminal 57 residues region is enough to interact with sub-domain II (RecA2) and sub-domain III (CTD) of the helicase.
Moreover, earlier studies which mapped the interaction between NS3 and NS4b used short fragments of NS4b fused with relatively larger fusion tags to fine map the interaction interface with NS3. The fusion tags (e.g. Sumo-tag) may interfere with the correct structural presentation of the NS4b fragment for interaction with NS3. For example, several studies (including those mentioned above) showed that NS4b exists as an oligomer inside the infected cell and in vitro (this is corroborated by our study results as well). However, the small peptides/short fragments of NS4b fused with expression tags (used in the above-mentioned studies) did not show such oligomerization behaviour. In our study, we found the NS3 helicase co-eluted with the oligomer population of NS4b full-length as well as NS4bN57 fragment, which is an indication that the proteins that we used are closer to their native conformation.
Another novel aspect of our study is that we used the 3’-stem-loop DNA/RNA oligo/s from the 3’-UTR of DENV genome - a native substrate for NS3 helicase during negative strand synthesis beginning. We also showed that NS4b interaction resulted in CTD (subdomain III) domain motion in NS3 helicase, which could explain the helicase activity enhancement upon NS3-NS4b interaction. To the best of our knowledge, this is the first experimental evidence of the DENV NS3 helicase mechanism, supporting the proposed mechanism made by other groups (references 33, 46 and 47 in our manuscript) based on the structural studies of DENV and other flavivirus NS3 helicases.
We request the editor and the reviewer to assess the novelty of our work in the context of the above-mentioned points. Our study was definitely guided and inspired by the study results that the reviewer mentioned in the comments. We believe that our study is a continuation and advancement of the work done by Zou et al. 2015, Chatel-Chaix et al., 2015 and Lu et al. 2021 to understand the NS3-NS4b interaction interface. We cited and thoroughly discussed all these studies in the Introduction and Discussion sections of our manuscript. Our thorough study, using various experimental and in silico approaches, conclusively showed that the N-terminal region of NS4b is essential for interaction with NS3, and characterized conformation dynamics in NS3 helicase subdomains upon interaction with NS4b, which explained the helicase activity modulation by NS4b.
Reviewer 2 Report
Authors may consider the following points and provide proper clarification.
1. Please specify whether periodic boundary conditions were applied for molecular dynamics experiment. Authors are also requested to mention the geometry of the grid utilized in the MD simulation.
2. In the RMSF plot (Figure 4D), both the 5’ UTR and 3’ UTR regions fluctuate. In addition, fluctuations at the 5’ UTR seems to be more compared to the 3’ UTR. Authors may justify their observation.
3. The structure of NS3 was predicted through homology modelling algorithm. On the other hand, the structure of NS4b was predicted using machine-learning algorithm (since no template was available). However, authors may generate the structure of NS3 using machine-learning approach and compare the two structures to get an idea about the structural difference (if any) arising out depending on two different methods used. Structural difference would have an impact on the downstream analysis.
4. According to Lu et al., 2021 the N terminal 51-83 residues of NS4B is important for its interaction with NS3. In the present study authors found that N terminal 1-57 resides are important for the interaction. In this regard it would be interesting to know whether the 7 residues that are common in both the studies are critical for this interaction? Authors could consider to make a construct of 1-50 N terminal residues of NS4B to address this question.
5. In figure 1C, the strength of interaction between full length NS4B with full length helicase NS3 is same when compared to its interaction with different domains of NS3. Which is quite surprising. the strength of interaction between full length NS4B with full length helicase NS3 should be higher when compared to interaction with different domains. If there is any explanation for this observation, authors should incorporate that in the manuscript.
Similarly, please explain why 1-57 N terminal residues of NS4B is showing more affinity towards NS3 helicase (full length and different domains) compared to full length NS4B.
Author Response
- Please specify whether periodic boundary conditions were applied for molecular dynamics experiment. Authors are also requested to mention the geometry of the grid utilized in the MD simulation.
Our response: we thank the reviewer for pointing to this inadvertent omission of important experimental detail about our MD studies in the manuscript. We included the details of periodic boundary conditions and the simulation box geometry (cubic box of 14.903 nm, 14.910 nm and 14.906 nm dimensions with Gromacs default parameter of 50 Å edge distance for periodic boundary conditions) used in the MD simulation in the revised manuscript. We incorporated these details at line 229 under the subsection 2.4 (paragraph 4) of Materials and Methods section.
- In the RMSF plot (Figure 4D), both the 5’ UTR and 3’ UTR regions fluctuate. In addition, fluctuations at the 5’ UTR seems to be more compared to the 3’ UTR. Authors may justify their observation.
Our response: I am afraid that the reviewer misread the Figure 4D. The reviewer may be referring to the subdomains I (N-terminal) and subdomain III (the CTD) of NS3. As we described in the Results section of the manuscript, (sub-section 3.4, paragraph 3) the RMSF plot is showing the residue wise root mean square deviation in the MD trajectory, comparing NS3 helicase alone model values to NS4b-NS3 docked model. As explained, The CTD-domain (subdomain-III of the helicase) region (orange boxed in the RMSF plot in Figure 4D) shows more fluctuation, indicating dynamic motion in the CTD upon NS4b interaction. As the reviewer rightly pointed out (assuming that the reviewer is pointing to the high RMSF differences in the sub-domain I region, along with the CTD region), subdomain I (referred to as RecA1 in our manuscript) also showed RMSF differences between NS3 alone and NS3-NS4b. As discussed in our manuscript (lines 749-754) earlier MD simulation studies predicted that RecA1 and RecA2 domains swivel over each other and show a ‘closing’ loop (p-loop) from RecA1 domain would move in after ATP-binding and hydrolysis. This dynamic change in conformation in the RecA1 and RecA2 domains is proposed as an allosteric mechanism to couple ATP-hydrolysis to RNA helicase activity (at the interface of CTD and RecA1-RecA2 domains). We did not use the ATP-bound NS3 model in our study. We observed conformational change in the RecA1 region in our MD simulation. It is possible that NS4b interaction may result in RecA1 subdomain conformational change as well (apart from the CTD domain motions). But, at this juncture, we will not be able to make any conclusions about the RecA1 subdomain motions upn NS4b interaction based on our our intrinsic fluorescence study results. We thank the reviewer for directing us to this interesting observation. We will pursue this in our continuing studies on NS3 helicase mechanism in future.
- The structure of NS3 was predicted through homology modelling algorithm. On the other hand, the structure of NS4b was predicted using machine-learning algorithm (since no template was available). However, authors may generate the structure of NS3 using machine-learning approach and compare the two structures to get an idea about the structural difference (if any) arising depending on two different methods used. Structural difference would have an impact on the downstream analysis.
Our response: We thank the reviewer for this suggestion. This helped us build more confidence in our NS4b structure modelling by AlphaFold 2.0. As suggested by the reviewer we modelled NS3 helicase structure with the DENV serotype 1 sequence using AlphaFold 2.0 (through a google colab notebook). AlphaFold 2.0 predicted structure and the structure model that was made using homology modelling (using SWISS-MODEL server) has Ca-RMSD value of 0.88 Å, indicating that both the structures were nearly identical. We described the results on modelling DENV serotype 1 NS3 using AlphaFold2.0 in the subsection 3.2, paragraph 3 of the Results section of the manuscript (line 566).
- According to Lu et al., 2021 the N terminal 51-83 residues of NS4B is important for its interaction with NS3. In the present study authors found that N terminal 1-57 resides are important for the interaction. In this regard it would be interesting to know whether the 7 residues that are common in both the studies are critical for this interaction? Authors could consider to make a construct of 1-50 N terminal residues of NS4B to address this question.
Our response: Following the reviewer’s suggestion we made a bacterial expression construct to express 1-50 N-terminal residues of NS4b. For reasons that we cannot understand the protein expression has become toxic to the bacterial cells, and we could not express and purify the protein. A probable explanation is, the first predicted helix, which would end at residue 57 of NS4b, might have misfolded due to truncation of the protein after residue 50. This might have resulted in toxicity to the bacterial cell. However, we take the reviewer’s suggestion and will pursue this further using different experimental methods in our future studies.
- In figure 1C, the strength of interaction between full length NS4B with full length helicase NS3 is same when compared to its interaction with different domains of NS3. Which is quite surprising. the strength of interaction between full length NS4B with full length helicase NS3 should be higher when compared to interaction with different domains. If there is any explanation for this observation, authors should incorporate that in the manuscript.
Our response: earlier studies by Zou et al (2015) showed that the NS4b interaction interface on NS3 maps to a broad region spanning subdomain II (RecA2) and subdomain III(CTD) of the NS3 helicase. As the reviewer rightly pointed out, it is expected that NS3 RecA2 & CTD and CTD alone constructs show a weaker interaction (less color intensity) compared to when full-length or RecA2&CTD NS3 constructs are used. Though we do not have an experimental validation, it is possible that having any one of the interacting domains (either RecA2 or CTD of NS3) is enough for interaction with NS4b. Alternatively, it is possible that the bacterial two-hybrid assay is not sensitive enough to show the interaction strength differences between different constructs of NS3 that we used, as long as some interaction is possible.
Similarly, please explain why 1-57 N terminal residues of NS4B is showing more affinity towards NS3 helicase (full length and different domains) compared to full-length NS4B.
Our response: we were equally surprised to see that NS4bN57, compared to the full-length protein, showed a more robust interaction with NS3 helicase. One possible reason is that most of the full-length protein (or N-terminal 57 residues truncated NS4b protein) made in the bacterial cell would have partitioned into the membrane (due to the TM helices in the C-terminus) and not available for interaction with NS3 in the cytosol. On the other hand, N-terminus 57 residues region would have been more readily available in the cytoplasm for interaction with NS3, giving a robust red color development.
Round 2
Reviewer 1 Report
The manuscript proposes a new model of membrane topology of NS4b (N-terminal region in the cytosolic side). It some contradicts previous reports Zou et al. (2015) and Chatel-Chaix et al. (2015). However, data from this study is insufficient to support this new model. It needs to strength evidences for NS4b N57 interact with NS3.
1. On Fig 1C, pUT18-NS4b (1-57) and pUT18-NS4b (1-249) should be co-transformation with a pKT25 plasmid expressing unrelated protein as negative control.
2. On Fig 3H, unrelated recombinant protein as negative control should be test interact with rNS3 to strength specificity of rNS3/NS4b interaction.
3. On Fig 3H, oligomerization of rNS4b could be change its density. How can you rule out oligomer effect on liposome/ rNS4b interaction assay.
4. On Fig 6C, rNS4B-N57 enhancing helicase activity of rNS3 is unclear.
Author Response
Author responses to reviewer comments
Manuscript Id and title: viruses-1799152; Dengue virus NS4b N-terminus Disordered Region Interacts with NS3 Helicase C-terminal Subdomain to Enhance the Helicase Activity
Round 2 Reviewer 1 comments:
The manuscript proposes a new model of membrane topology of NS4b (N-terminal region in the cytosolic side). It some contradicts previous reports Zou et al. (2015) and Chatel-Chaix et al. (2015). However, data from this study is insufficient to support this new model. It needs to strength evidences for NS4b N57 interact with NS3.
Our response:
We thoroughly discussed this aspect in our manuscript (both in the Introduction section and the Discussion section, lines 1000-1034), clearly explaining our reasoning/rationale for proposing an alternative model of membrane topology of NS4b - different from what was proposed by Miller et al., (2006) (reference 29 in our manuscript).
Miller et al. (2006), based on their studies, also discussed the possibility of a topology for NS4b, flipped to the one that they proposed in their publication (paragraphs 5 and 6 in the Discussion section of the article, Miller et al. 2006, JBC vol. 281: page 8854-8863). Subsequently, the following published studies also strongly indicate that the N-terminus of NS4b is probably on the cytosol side of the RO: i. Zou et al. (2014)[19] based on their fluorescence protease protection assay results with DENV 2K-NS4b(1-93)-EGFP, inferred that the N-terminus 100 residues may position on ER lumen side as well as cytosol side of the RO, ii. Zou et al. (2015)[reference 24 in our mss] could immuno-precipitate NS4b from DENV-infected cell lysates without detergent extraction of the protein. Since majority of the NS4b, except for the N-terminal 100 amino acids, insert into the membrane, the only explanation of this result is that the N-terminus is in the cytosolic side of the RC, where NS3 is also present; iii. The N-terminal 95 residues of NS4b are required for modulation of IFN-gamma signaling[reference 31 in our mss] in DENV infected cells, probably through its interaction with STAT-1 protein – a cytosolic protein (these points are discussed in our manuscript).
Thus our proposal that the N-terminal region of NS4b is in the cytosolic side of RO (as depicted in the membrane topology for NS4b in Figure 1 of our manuscript) is not unfounded. And, all our biochemical studies and that of Lu et al., 2021 (Virus Research, 2021) do corroborate our proposal.
- On Fig 1C, pUT18-NS4b (1-57) and pUT18-NS4b (1-249) should be co-transformation with a pKT25 plasmid expressing unrelated protein as negative control.
Our response: We request the referee to refer to the Figure 1C subpanel, labelled negative control. Our bacterial two-hybrid experiments did include a negative control and results from the negative control experiment are already included in the Figure 1C. As can be seen from the Figure 1C (under the subpanel heading ‘negative control’, there is no color development when only pUT18 and pKT25 vectors are used for co-transformation of the bacterial cells. This served as a negative control, whereas a canonical interacting protein subdomain (Leucine zipper protein) fusion to T25 and T18 resulted in red color development (positive control).
- On Fig 3H, unrelated recombinant protein as negative control should be test interact with rNS3 to strength specificity of rNS3/NS4b interaction.
Our response: We tested interaction between NS4b and NS3 using recombinant purified proteins. Hence, the only possible explanation for the co-elution of rNS3 with rNS4b oligomeric population is a specific and stable interaction between the proteins. The specific interaction is further corroborated by increasing amounts of rNS3 seen co-eluting with NS4b when higher stoichiometries of NS3:NS4b (1:3 and 2:3) are used in the experiment.
We respectfully disagree with the reviewer about the experiment suggested (to test the interaction between unrelated recombinant protein and rNS3) – this may not be a conclusive experiment to test the interaction between NS3 and NS4b.
Moreover, through these experiments, we are merely corroborating, biochemically, what has been thoroughly established about NS3 and NS4b interaction using cell-based (co-IP, co-localization), and viral infection studies.
- On Fig 3H, oligomerization of rNS4b could be change its density. How can you rule out oligomer effect on liposome/ rNS4b interaction assay.
Our response: The reviewer is probably referring to Figure 3I, where liposome-rNS4b association is studied using liposome co-floatation studies.
We use liposome co-floatation assays in our lab regularly to test association of viral proteins with liposomes. We never observed sedimentation of liposomes due to increased density after a viral protein association, as the reviewer was suspecting. The lipid LUVs that we used in our study are 100 nm in size (average diameter). And, even assuming insertion of several molecules of rNS4b into the LUV membrane would not change the density that it will sediment to the bottom in the density gradient.
Thus, the only possible explanation for the rNS4b detected in the bottom fractions of the tube after density gradient centrifugation is that the protein did not associate and co-float with the liposomes. We confirmed that the liposomes are intact and in the upper fractions of the tube by diphenyl hexatriene probing.
- On Fig 6C, rNS4B-N57 enhancing helicase activity of rNS3 is unclear.
Our response: As the reviewer correctly pointed out, the extent of increase in helicase activity of rNS3 upon interaction with NS4bN57 is not as robust as that seen with full-length rNS4b. (this result is described as such in our manuscript, lines 920-925). As we explained, It is possible that though the N-terminus 57 residues region of NS4b is enough to interact with NS3 helicase, other regions of the NS4b may be important for the helicase activity modulation in ways that we cannot explain from our study.